# Evaluating and improving the representation of bacterial contents in long-read metagenome assemblies

Xiaowen Feng[1,2] and Heng Li[1,2*]

---

*Correspondence:
hli@ds.dfci.harvard.edu

[1] Department of Data Sciences,
Dana-Farber Cancer Institute,
Boston, USA
[2] Department of Biomedical
Informatics, Harvard Medical
School, Boston, USA

## Abstract

**Background:** In the metagenomic assembly of a microbial community, abundant species are often thought to assemble well given their deeper sequencing coverage. This conjuncture is rarely tested or evaluated in practice. We often do not know how many abundant species are missing and do not have an approach to recover them.

**Results:** Here, we propose *k*-mer based and 16S RNA based methods to measure the completeness of metagenome assembly. We show that even with PacBio high-fidelity (HiFi) reads, abundant species are often not assembled, as high strain diversity may lead to fragmented contigs. We develop a novel reference-free algorithm to recover abundant metagenome-assembled genomes (MAGs) by identifying circular assembly subgraphs. Complemented with a reference-free genome binning heuristics based on dimension reduction, the proposed method rescues many abundant species that would be missing with existing methods and produces competitive results compared to those state-of-the-art binners in terms of total number of near-complete genome bins.

**Conclusions:** Our work emphasizes the importance of metagenome completeness, which has often been overlooked. Our algorithm generates more circular MAGs and moves a step closer to the complete representation of microbial communities.

**Keywords:** Metagenome, Binning, Metagenome-assembled genomes, Assembly completeness

## Background

De novo metagenome sequencing promises unbiased and comprehensive snapshot of microbial communities of interest, independent of isolation and cultivation [1–3]. Ideally we would like to reconstruct circular genomes for each species and strains. The reality is not as encouraging.

Past metagenome sequencing projects focused on short read sequencing. They could not reconstruct circular genomes automatically due to repeat contents or strain similarity. Instead, these projects produced short contigs of tens of kilobases (kb) in length, ~100 times shorter than a typical bacterial genome. To obtain a more complete representation of an individual species, we typically apply binning algorithms to group short contigs into metagenome-assembled genomes (MAGs). The completeness of a MAG is often measured with CheckM [4] and analysis often focus on near-complete MAGs [5] in downstream analysis. A MAG from short-read assembly, even if near-complete, is fragmented and could be composed of segments from multiple related strains. It is not equivalent to a whole genome.

The advent of long-read sequencing technologies combined with specialized long-read metagenome assemblers [6–8] has dramatically improved the completeness of individual MAGs, with MAG yield per unit library size similar to that of short read sequencing projects (Additional file 7: Figs. S1 and S2). It is now possible to assemble tens to hundreds of circular genomes from one deeply sequenced metagenome sample. Nonetheless, even with long reads, many species are still unresolved and binning is still necessary. While binners initially developed for short reads work for long-read metagenome contigs, they have not considered the capability of long-read assemblers to separate closely related strains and sometimes group circular contigs together, which results in highly contaminated bins [7]. They do not work well straight out of box. In addition, as we will explain later, current binners, including graph-based binners [9, 10], are unable to capture the circular topologies in accurate long-read assembly graphs. There is still room for improvement.

When we could routinely obtain continuous, complete bacterial genomes in metagenome samples, we may start to ask how well these MAGs represent a given sample as a whole. A comprehensive answer to this question poses its own challenges. We could map reads back to the assembly and check how many reads are unmapped. This unfortunately does not tell us the characteristics of under-represented species and thus does not inform how to improve our assemblies. It may be tempting to think abundant species could be well reconstructed [11–13], but this is often not the case for practical samples [14, 15]. Horizontal gene transfer (HGT) or large duplication events may be even more difficult to resolve. We lack the necessary tools to evaluate the representation completeness of a metagenome assembly.

In this article, we proposed two methods to investigate the representation of bacterial contents in long-read metagenome assemblies. The first method is inspired by KAT [16] and Merqury [17]. It evaluates the completeness at different *k*-mer occurrence thresholds. The second method leverages the observation that long reads are longer than 16S RNA. It checks if 16S RNAs are captured by assemblies. Both methods confirm that abundant species may not always be assembled. To improve the representation completeness, we additionally developed a new binning algorithm that finds circular paths in the assembly graph. The algorithm itself is comparable to other binning algorithms in accuracy and can be used together with other algorithms for higher binning quality.

## Results

### Overview of the hifiasm-meta binning algorithm

In the hifiasm-meta assembly graph (Fig. 1A), we noticed circular unresolved subgraphs which are likely due to high strain diversity or shared homology around HGT. Checking the CheckM1 report, we found these circles often corresponded to real bacterial genomes. The observation inspired us to develop an algorithm to enumerate circular paths in the assembly graph. More exactly, we perform a series of depth-first search (DFS) traversals on the assembly graph in attempt to find circular paths of roughly 500 kb or longer. Some circular paths found this way may come from one subgraph and may be redundant. We apply the Mash MinHash-based algorithm [18] to estimate the pairwise distances between circular paths and remove a circular path if it is ≥95% similar to another circular path. This procedure gives us a list of non-redundant circular paths.

Circle rescue cannot group disconnected contigs and thus does not replace classical binning. To provide a smoother user experience, we also implemented a native binning algorithm tightly integrated in hifiasm-meta. At this step, we ignore circular contigs or linear contigs that are on non-redundant circular paths. For each remaining contig, we

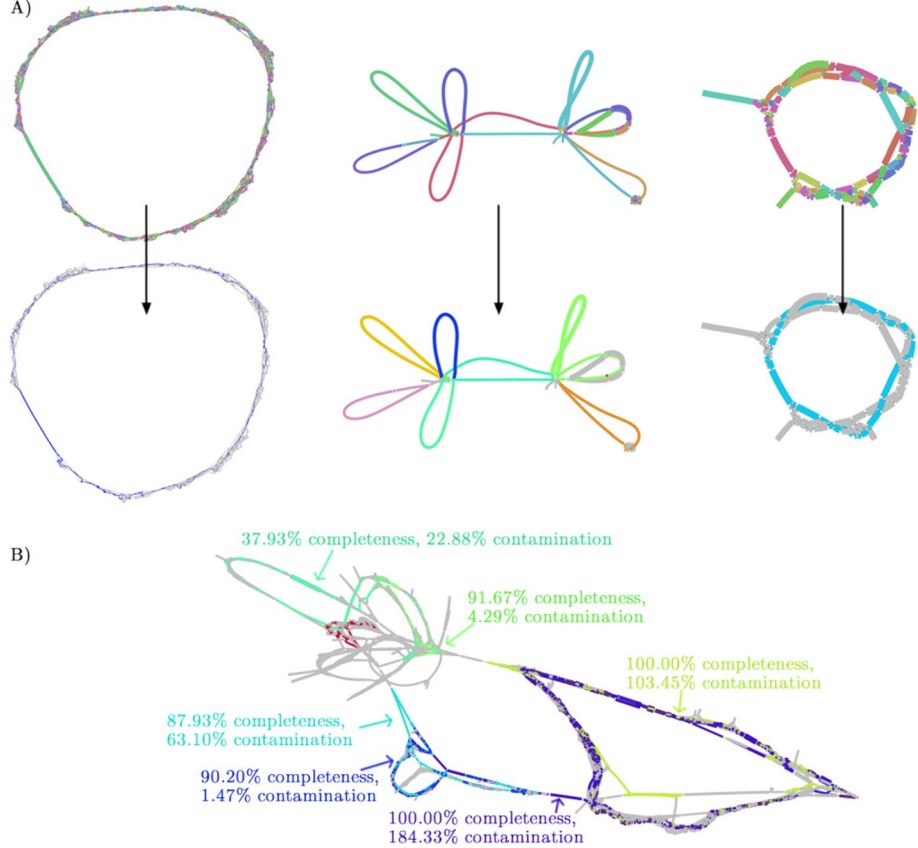

**Fig. 1** Showcases of when the circle rescue works well and not well. **A** The top row shows visualization of three subgraphs from hifiasm-meta primary assemblies, with contigs colored randomly [19]. The bottom row shows the circles (i.e., putative MAGs) rescued by the heuristics, with contigs of each circle colored the same. Contigs that do not belong to any rescued circle are colored gray. These circles were near-complete MAGs based on CheckM1. **B** The heuristics does not work well in subgraphs or regions like this one. In this particular example, the contigs had very low coverages

compute the read coverage in the logarithm scale, count the occurrences of canonical 5-mers and then create a feature matrix of size $N \times (512 + 1)$, where $N$ is the number of considered contigs, 512 is the number of canonical 5-mers, and the last dimension is for read coverage. We normalize each column in the matrix by $Z$-score, embed it to a 2-dimensional space with t-SNE, and extract contig bins in the embedded space using a small radius as follows. We seed a cluster from the longest non-circular contigs to the shortest. For each seed, we try to find a circle with diameter $D$ on the t-SNE plane such that it contains the seed and all other contigs that are $\leq D$ away from the seed. These contigs are called neighbors. If found, the seed and neighbors are put into a bin and marked as used; otherwise, we test whether the circle centered at the seed with diameter $1.6 \times D$ contains all neighbors and create a bin on success. When both attempts fail, we record the seed as a single-contig bin if it is longer than 500 Kb; otherwise, we do nothing. Finally, we merge the resultant bins with circular contigs and non-redundant circular paths to generate the final results.

We call the algorithm above as *hmBin* in this article. Similar to vamb, hmBin only uses information from the metagenome reads without using marker genes or relying on known reference bacterial genomes. Nonetheless, hmBin still has a few hyperparameters such as the minimum length of circular paths, the Mash similarity, and the radius used for defining t-SNE clusters. During the development of hmBin, we explored different thresholds and found the binning results are generally insensitive to these thresholds on our datasets ("Methods"). We also tried to use UMAP for clustering. The result was slightly worse than the t-SNE clustering.

The circle rescue heuristic also works with mdBG [20] assembly graphs and can recover a third to a fourth of near-complete MAGs found by hifiasm-meta in tested samples. However, mdBG does not correct insertion/deletion sequencing errors in homopolymers which often interrupt open reading frames and lead to low CheckM1 completeness. We thus did not evaluate mdBG results. MetaFlye does not produce an assembly graph for contigs. It instead generates a repeat graph which, unlike string graphs used by hifiasm-meta, may reuse a unitig in multiple contigs. Our circle finding algorithm only traverses each unitig or contig once and does not work with metaFlye assembly graphs. In theory, it is possible to adapt our algorithm for repeat graphs. However, metaFlye assembly graphs more often look like the one in Fig. 1B. We would not be able to get good results anyway.

### Understanding the behavior of the hifiasm-meta binning algorithm

We collected 19 HiFi datasets (Additional files 1 and 2: Tables S1 and S2) and evaluated the quality of MAGs with CheckM1 [4]. Following previous work [21–23], we use the following criteria to evaluate MAGs. "Near-complete" means $\geq 90\%$ completeness and $< 5\%$ contamination. "High-quality" means $\geq 70\%$ completeness, $< 10\%$ contamination but does not qualify for near-complete. "Medium-quality" means $\geq 50\%$ completeness, $\geq 50$ quality score but does not qualify for the above two where the quality score of a MAG is defined as "completeness $- 5 \times$ contamination." All other MAGs are referred to as failed-quality. The categories never refer to the original MiMAG definition, with or without being suffixed by "MAG," e.g., "HQ" means the high-quality category or the high-quality MAGs under our definition.

Figure 2 shows the flow of data when we assembled the sheep-gut-1b dataset with hifiasm-meta and evaluated the results with CheckM1. Notably, a small fraction of circular contigs are not considered near-complete. They may come from underrepresented clades in the CheckM1 data as we found earlier [7]. On sheep-gut-1b, hmBin recovered tens of circular contigs that are considered near-complete by CheckM1 (Fig. 3).

### Optimizing other binning algorithms for hifiasm-meta assembly

We additionally applied four other binning algorithms to our datasets, including Vamb [24], MetaBAT2 [25], GraphMB [10], and SemiBin1 [26]. They use different information and distinct algorithms. Vamb only considers coverage and tetranucleotide profiles for binning. MetaBAT2, possibly the most widely used binning algorithm, additionally trains hyperparameters on existing bacterial genomes. GraphMB takes graph topology into account with a Graph Neural Network. SemiBin1 is special in that it uses single-copy marker genes to guide binning, the same information CheckM1 uses to evaluate MAGs. It may be biased to known species and will be unfairly favored by CheckM1. GraphMB can optionally use single-copy marker genes as well. We did not evaluate that mode. We also note that some binners, such as vamb, are optimized for jointly binning multiple samples and may underperform given a single sample.

As hifiasm-meta may assemble strains of the same species into separate contigs, Vamb, MetaBAT2, and GraphMB may group these contigs and result in complete but highly contaminated bins. To address this issue, we post-process the bins from these binners by putting ≥ 1 Mb circular contigs into separate single-contig bins. Figure 4A shows the effect of putting circular contigs into separate bins. This post-processing step greatly increases the number of near-complete MAGs for the three binners. The improvement is especially notable for the two sheep gut datasets. This step does not help SemiBin1, probably because SemiBin1 can use single-copy marker genes to identify this problem while binning.

Meanwhile, as the hifiasm-meta circle rescue heuristic and traditional binners use different information for binning, the circular paths hifiasm-meta identifies may not always be captured by post-processed bins. We thus additionally merge the circular paths with post-processed bins as follows. For a bin found by a traditional binner of 0.5–10 Mb in size, we discard it as a redundancy if the bin has > 1 Mb sequences or > 10 contigs shared

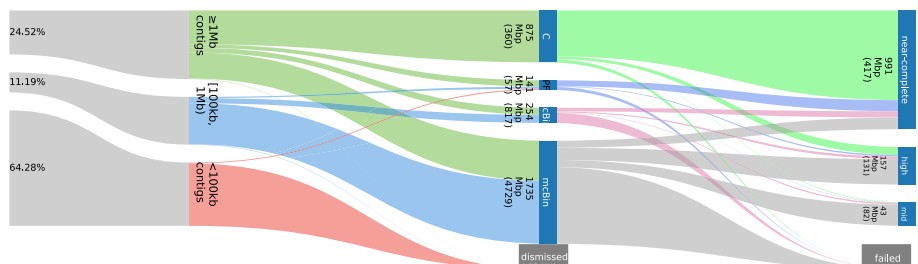

**Fig. 2** Sankey plot showing flow of reads and contigs of sheep-gut-1b. Left: reads to contigs, categorized by contig length. Middle: contigs to binning categories (from top to bottom, "C" for ≥ 1Mb circular contigs; "PF" for circular path rescue; "scBin: single contig bin; "mcBin" for multi-contig bin; dismissed: unused by binning). Right: MAGs quality categories. Group heights are normalized by counts, except the left-most side which is normalized by base pairs. For the visualization purpose, the height of "dismissed" and the height of "failed" are not proprotional to the counts

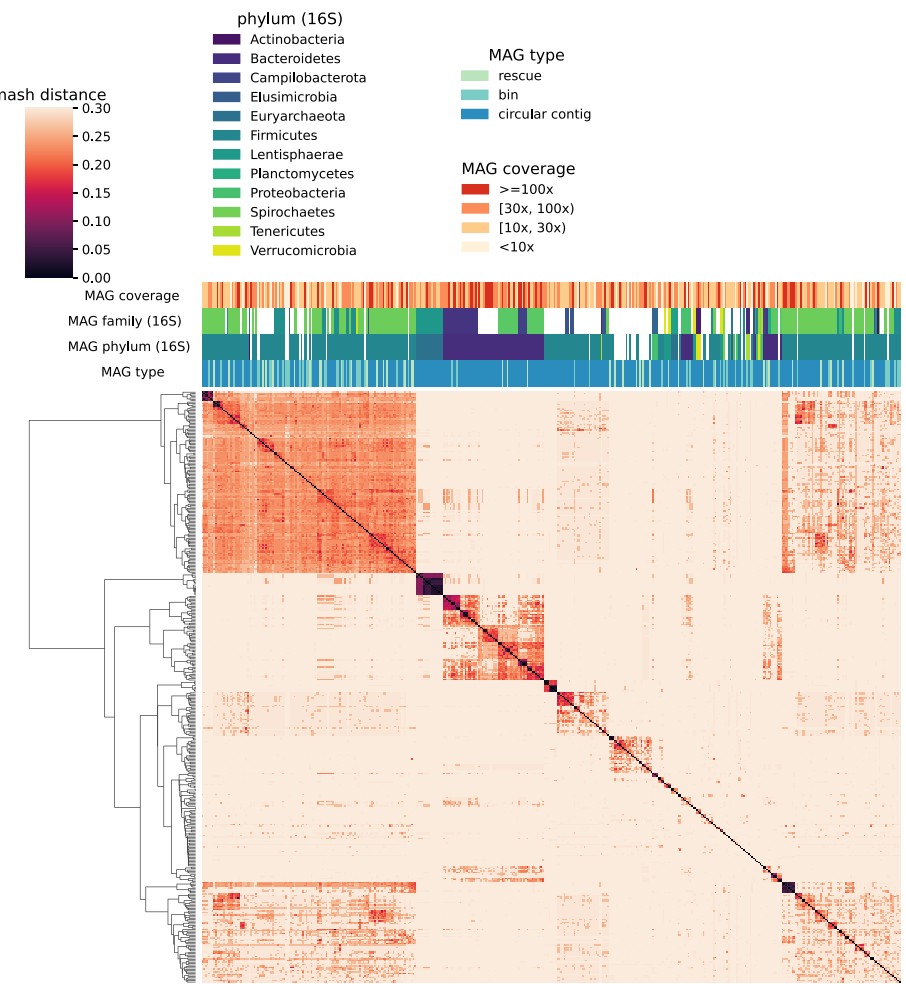

**Fig. 3** Clustering near-complete MAGs of sheep-gut-1b. The middle-left dense darker square represents Archaea. Mash distance more than 30% are shown as 30%

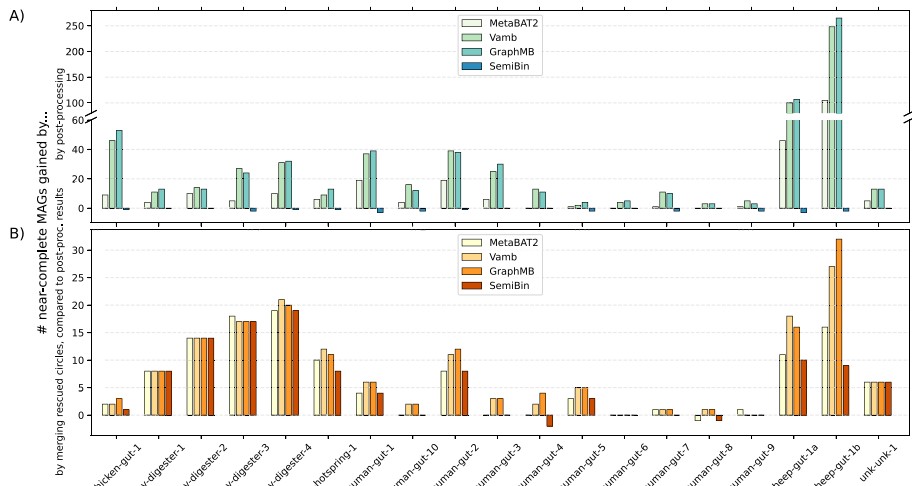

**Fig. 4** Effect of post-processing and circle rescue on binning quality. **A** Number of additional near-complete MAGs recovered by putting ≥ 1 Mb circular contigs to separate bins. **B** Number of additional near-complete MAGs recovered by the hifiasm-meta circle rescue heuristic

with a rescued circular path. We then combine circular contigs, rescued circles and the remaining bins to produce the final binning results.

Across all the tested datasets, 74 near-complete MetaBAT2 bins were discarded by the procedure above. Sixty-four of them have < 1% mash distance to rescued circles (52 of them < 0.1%). In 61 out of the 64 cases, rescued circles were better than the rejected MetaBAT2 bins in terms of equal or better CheckM1 completeness and contamination. In the remaining three cases, the CheckM1 report on the rescued circles are all close to the report on the MetaBAT2 MAGs (100.0/0.0%, 99.33/1.34% and 90.7/0.94% for MetaBAT2 MAGs, and 98.8/0.48%, 97.32/1.34% and 99.06/2.96% for rescued circles, respectively).

Conversely, 79 out of the 193 near-complete rescued circles were not within 5% mash distance from any near-complete MetaBAT2 bins. Our circle rescue strategy found additional near-complete MAGs on top of MetaBAT2. It also improved Vamb, GraphMB, and SemiBin1 (Fig. 4B), suggesting our strategy captures additional information missed by other binners.

On the binning performance, the integrated hifiasm-meta binning algorithm, hmBin, finds more near-complete MAGs than the raw output of MetaBAT2, Vamb, and GraphMB (Table 1; Additional files 3 and 6: Tables S3 and S6). It is broadly comparable to optimized Vamb and GraphMB and rivals MetaBAT2 only with the post-processing step. The optimized MetaBAT2 with rescued circles performs better than hmBin especially on the two sheep gut datasets. SemiBin1 gives the most number of near-complete MAGs overall, potentially because it uses the same information as CheckM1 during binning.

### Evaluating the representation completeness with *k*-mer spectra

Inspired by KAT [16] and mercury [17], we use *k*-mers to evaluate the completeness and redundancy of a metagenome assembly. Let $M_x^{(c)}$ be the count of *k*-mers that occur $x$ times in reads and $c$ times in the assembly. KAT plots $M_x^{(0)}$, $M_x^{(1)}$, $M_x^{(2)}$, etc. Because in a metagenome assembly, *k*-mer counts are affected by the abundance of genomes and are highly variable, the KAT plot is hard to read. To address this issue, we plot the fraction of *k*-mers instead. More exactly, let $N_x^{(c)} = \sum_{y=x}^{\infty} M_y^{(c)}$ be the number of *k*-mers occurring $\geq x$ times in reads and exactly $c$ times in the assembly, and let $N_x = \sum_c N_x^{(c)}$ be the number of *k*-mers occurring $\geq x$ times in reads. We stack $N_x^{(c)}/N_x$ of different $c$ and plot them together.

In such a plot (Fig. 5; Additional file 7: Figs. S3, S4 and S5), each number $c$ occupies a band, which we call as the "$c\times$ band." The blue area, for example, corresponds to the $0\times$ band, which represents unassembled read *k*-mers. We expect to see a high $0\times$ band at $x = 1$ due to sequencing errors in reads. For a complete assembly, the blue $0\times$ band should be close to 0 at $x > 15$, a typical read depth at which assemblers start to produce contiguous contigs. Given a metagenome sample composed of genomes with distinct sequences, a complete non-redundant assembly representing the sample will ideally have the $1\times$ band spanning the entire plot up to the read depth of the most abundant genome. Sample "chicken-gut-1" is such an example. On real data, hifiasm-meta may be able to resolve similar strains into separate contigs. *k*-mers from these

**Table 1** CheckM1 evaluation of binning algorithms. The second column shows the $N_d$ diversities estimated by Nonpareil [27], which is empirically correlated with alpha diversity. The third column shows the number of circular near-complete contigs, near-complete contigs (≥ 90% completeness and < 5% contamination), and high-quality contigs (≥ 70% completeness and < 10% contamination) before binning. The three numbers in each following cell give the number of near-complete MAGs, high-quality MAGs, and medium-quality MAGs, respectively. In the table, "raw" stands for raw output by the binning algorithm; "+post" for post-processing by putting ≥ 1Mb circular contigs into separate bins; "+rescue" for merging with circular paths rescued based on the graph topology. Column "All" shows the count of near-complete MAGs in a union of all binners (deduplicated at 1% mash distance), and column "hmBin unique" shows the count of near-complete MAGs SemiBin1 was run in the long-read mode and GraphMB was run without knowledge of single-copy marker genes

| Dataset | $N_d$ diversity | Circular complete contigs, complete contigs, high-quality contigs | hmBin (raw) | MetaBAT2 (raw) | MetaBAT2 (+post) | MetaBAT2 (+post)(+rescue) | Vamb (+post)(+rescue) | GraphMB (+post)(+rescue) | SemiBin1 (raw) | SemiBin1 (+rescue) | All (+post)(+rescue) | hmBin unique |
|---|---|---|---|---|---|---|---|---|---|---|---|---|
| chicken-gut-1 | 18.29 | 74\|78\|17 | 93\|21\|18 | 81\|27\|16 | 90\|32\|16 | 92\|30\|16 | 88\|24\|16 | 81\|15\|14 | 95\|40\|27 | | 99 | 1 |
| env-digester-1 | 18.98 | 20\|22\|14 | 33\|20\|19 | 23\|16\|16 | 27\|19\|15 | 35\|21\|18 | 32\|21\|17 | 34\|24\|19 | 35\|34\|25 | | 38 | 8 |
| env-digester-2 | 18.59 | 29\|38\|27 | 55\|23\|24 | 34\|30\|28 | 44\|31\|29 | 58\|35\|31 | 50\|40\|27 | 52\|38\|29 | 58\|62\|47 | | 62 | 14 |
| env-digester-3 | 18.31 | 39\|47\|21 | 67\|38\|32 | 48\|40\|31 | 53\|42\|33 | 71\|44\|35 | 70\|46\|36 | 69\|43\|38 | 76\|72\|52 | | 76 | 17 |
| env-digester-4 | 18.57 | 44\|58\|25 | 85\|29\|29 | 58\|38\|29 | 68\|37\|31 | 87\|34\|33 | 79\|33\|22 | 81\|28\|28 | 95\|58\|43 | | 100 | 17 |
| env-hotspring-1 | 18.45 | 16\|19\|11 | 33\|14\|14 | 18\|22\|13 | 24\|21\|14 | 34\|21\|14 | 33\|19\|18 | 33\|16\|15 | 35\|28\|21 | | 37 | 10 |
| human-gut-1 | 19.20 | 51\|68\|30 | 80\|29\|18 | 64\|35\|25 | 83\|41\|28 | 87\|41\|29 | 79\|36\|20 | 79\|37\|19 | 93\|61\|64 | | 104 | 4 |
| human-gut-10 | 18.44 | 25\|34\|14 | 39\|18\|16 | 40\|23\|22 | 44\|24\|22 | 44\|24\|24 | 40\|17\|20 | 38\|17\|18 | 42\|31\|32 | | 50 | 2 |
| human-gut-2 | 19.17 | 59\|73\|34 | 84\|34\|24 | 66\|41\|30 | 85\|47\|32 | 93\|48\|32 | 84\|44\|42 | 84\|41\|41 | 94\|69\|73 | | 103 | 5 |
| human-gut-3 | 17.31 | 42\|49\|1 | 52\|4\|9 | 45\|10\|12 | 51\|10\|12 | 51\|10\|12 | 53\|9\|7 | 51\|9\|6 | 53\|12\|13 | | 56 | 0 |
| human-gut-4 | 19.00 | 19\|25\|9 | 33\|18\|13 | 38\|29\|33 | 38\|31\|33 | 38\|30\|31 | 34\|27\|26 | 38\|22\|25 | 42\|39\|36 | | 48 | 0 |
| human-gut-5 | 18.53 | 7\|10\|12 | 19\|8\|7 | 18\|9\|10 | 19\|11\|11 | 22\|10\|11 | 19\|16\|9 | 15\|13\|12 | 23\|17\|19 | | 24 | 2 |
| human-gut-6 | 16.99 | 7\|11\|4 | 14\|7\|7 | 15\|14\|9 | 15\|14\|9 | 15\|13\|9 | 13\|8\|6 | 13\|8\|6 | 17\|12\|10 | | 18 | 0 |
| human-gut-7 | 18.51 | 17\|22\|14 | 24\|15\|13 | 25\|21\|20 | 26\|22\|21 | 27\|22\|21 | 22\|19\|14 | 24\|17\|15 | 26\|30\|22 | | 28 | 0 |
| human-gut-8 | 17.75 | 3\|6\|3 | 11\|8\|3 | 19\|13\|15 | 19\|13\|15 | 18\|13\|15 | 7\|8\|10 | 8\|5\|10 | 15\|15\|19 | | 22 | 0 |
| human-gut-9 | 17.87 | 12\|15\|7 | 25\|5\|5 | 22\|13\|21 | 23\|13\|21 | 24\|13\|21 | 24\|12\|15 | 22\|9\|19 | 29\|10\|14 | | 30 | 0 |
| sheep-gut-1a | 19.44 | 144\|186\|40 | 189\|53\|28 | 147\|52\|30 | 193\|62\|32 | 204\|60\|30 | 188\|56\|31 | 189\|50\|32 | 212\|80\|57 | | 217 | 9 |
| sheep-gut-1b | 19.53 | 317\|423\|100 | 417\|131\|82 | 345\|116\|94 | 450\|134\|100 | 466\|136\|96 | 414\|130\|75 | 403\|109\|56 | 490\|211\|167 | | 509 | 7 |
| unk-unk-1 | 19.17 | 25\|28\|12 | 34\|21\|11 | 25\|20\|16 | 30\|20\|17 | 36\|23\|18 | 37\|24\|16 | 37\|25\|17 | 40\|36\|21 | | 41 | 7 |

contigs would occur twice or more in contigs and would not be shown in the $1\times$ band (e.g., "human-gut-2" in Fig. 5).

From the $k$-mer spectrum plots, we found that long circular contigs alone do not provide a complete view of the corresponding libraries in most samples (Fig. 5, left column). Binning helped to improve the $k$-mer coverage, although the magnitude of the improvement varied between libraries (Fig. 5, right column). In three libraries (chicken-gut-1, sheep-gut-1a and sheep-gut-1b), merged MAG were close to the near-ideal situation demonstrated in Fig. S5 (Additional file 7).

To understand the content of $k$-mers in the $0\times$ band, we extracted these $k$-mers and aligned them to MAGs with bwa aln [28] to see if they can be found by allowing a few mismatches or indels. This only had minor effect on the plot. To check the $k$-mers content in the $2\times$ or higher bands, we extracted human-gut-10 $k$-mers in these bands that occurred > 800 times in reads, manually examined a few $k$-mers with their flanking regions, and found them to harbor ubiquitous genes (e.g., tRNAs) or horizontally transferable sequences (see the "Methods" section).

### Evaluating the representation completeness with 16S rRNAs

As most HiFi reads are a few times longer than 16S rRNA genes, we can estimate the composition of 16S rRNA directly from reads without assembly. Predicting the rRNA genes is well-studied [29, 30]. 16S-based taxonomy annotation has been similarly extensively explored [29, 31, 32], though the annotation accuracy may vary with the existing reference data. For example, out of 1.8 million reads from human-gut-9, 1.4% were identified to contain 16S rRNA and 90% of 16S reads were assigned to the genus level confidently. In contrast, out of 1.0 million reads from env-digester-1, 1.3% were identified to contain 16S but only 21% of 16S reads were assigned to the genus level. Non-human gut samples were something in-between: in sheep-gut-1b, 1.2% of reads contained 16S, with 47.3% of them having confident genus-level annotation. Due to this large differences between datasets, we do not trust the the species- or genus-level annotation of 16S from existing tools [33, 34].

To evaluate if MAGs could recover most 16S RNAs, we identify 16S RNAs from reads and greedily cluster them into OTUs such that 16S in an OTU is > 99% in identity [33] ("Methods"). No assembly could recover all abundant OTUs, but those

(See figure on next page.)

**Fig. 5** *K*-mer spectra of all samples. Given input reads and a set of contigs assembled from the reads, let $N_x^{(c)}$ be the number of $k$-mers occurring $\geq x$ times in reads and exactly $c$ times in contigs (thus "right-accumulated"). Then, $N_x = \sum_c N_x^{(c)}$ is the number of $k$-mers occurring $\geq x$ times in reads. Plots on the left column show the $k$-mer spectra of $\geq 1$ Mb circular contigs. The height (i.e., the length on the *Y*-axis) of the blue area intersecting at $x$ equals $N_x^{(0)}/N_x$. It is the fraction of read $k$-mers occuring $\geq x$ times in reads but absent from the assembly. The height of the orange area at $x$ equals $N_x^{(1)}/N_x$. The white dashed line shows $N_x$ and each plot is truncated at $x_t$ where $N_{x_t} < 10^6$. Intuitively speaking, a large blue region in the right part of a plot suggests an incomplete assembly that misses many high-abundance $k$-mers in reads. Plots on the right column show the $k$-mer spectra of hmBin MAGs. The light orange area with forward hatches indicates the contribution of rescued circles and the brown area with backward hatches indicates the contribution of MAGs found by non-topology based binning. The blue area is generally smaller due to the inclusion of non-circular MAGs

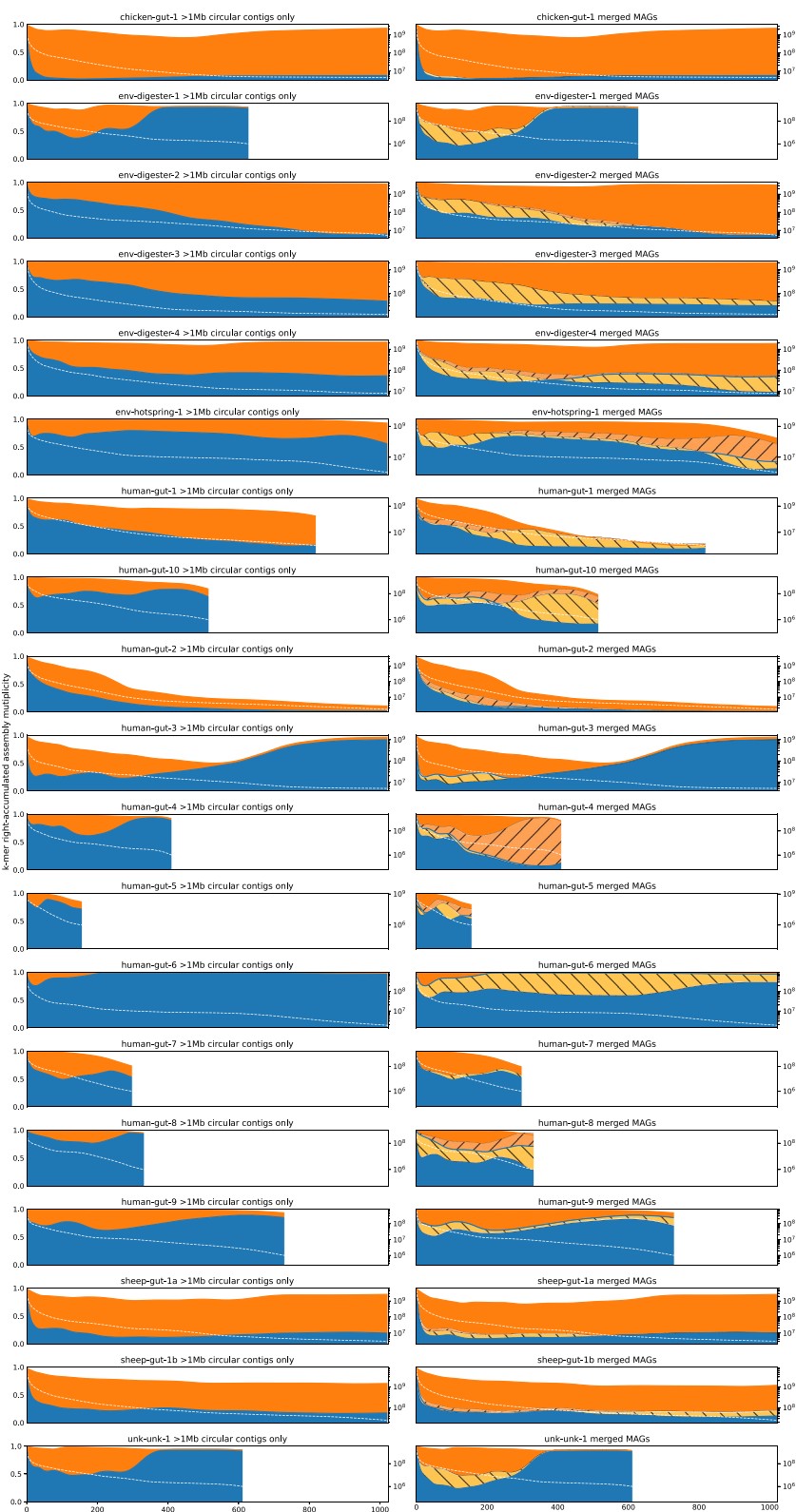

**Fig. 5** (See legend on previous page.)

evaluated to be better in *k*-mer spectrum approach missed less (Fig. 6; Fig. S6 (Additional file 7) provides plots of all samples as well as two additional OTU boundaries).

### Discovery of species unseen in catalogs

Despite the limited number of human gut samples used in this study, we still found MAGs that were absent from the existing human gut catalogs at species level (under mash distance ≤5%). Of our near-complete MAGs reconstructed from eight individuals and two 4-pooled libraries, 92/381 (24%) were not found among near-complete MAGs from Almeida et al., and 29/381 (8%) were not found in RefSeq. We did not find significant coverage difference between novel MAGs and known MAGs. We also compared the near-complete MAGs from chicken-gut-1 to the ICRGGC catalog [35] which consists of 12339 MAGs derived from 799 samples and found nearly all of hifiasm-meta MAGs have matches (85/93 at the species level and additional 7/93 at the strain-level). PRJNA657473 gives a catalog of ruminant gut community which consists of 10371 MAGs derived from 370 specimen sampled from seven species across ten gastrointestinal tract regions. The sheep samples in our study are very different. Out of the 490 near-complete MAGs in sheep-gut-1b, only one MAG matches a known MAG from PRJNA657473 at the species level. Overall, while the catalog and collections of reference assemblies have accumulated a large amount of samples, de novo assembly of HiFi reads is valuable, especially for some non-human samples.

### Discussions

We have improved the assembly quality of hifiasm-meta since its publication [7]. On top of these changes, we have implemented a binning algorithm tightly integrated into hifiasm-meta. This binning algorithm, hmBin, has two components: one to identify and rescue circular paths in the assembly graph and the other to cluster disconnected contigs using t-SNE embedding. The first component can also be combined with other binning algorithms. We showed that our algorithm led to more complete representation of

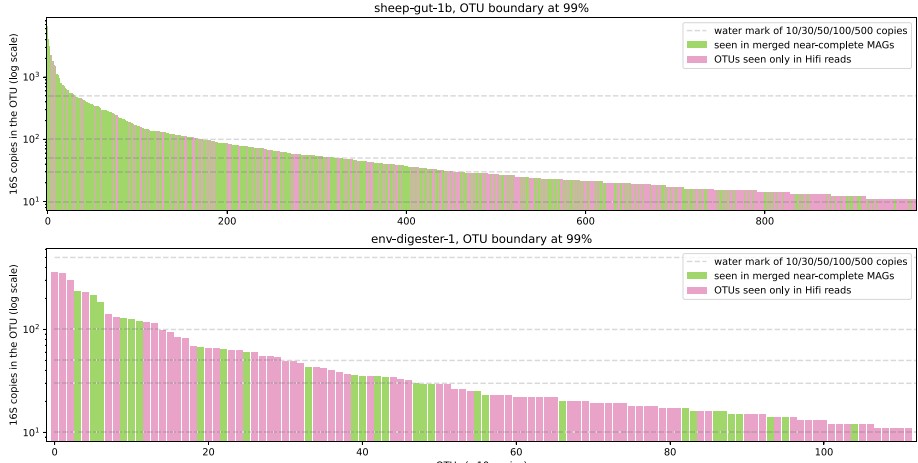

**Fig. 6** OTU recovery of merged bins, showing sheep-gut-1b and env-digester-1, which evaluated good and poorly in *k*-mer spectrum plot, respectively

bacterial populations in metagenome samples instead of fragmented contigs as shown in Fig. 1A.

Because the circle rescue heuristic only looks for circular paths, it will naturally miss linear chromosomes in small Eukaryotic genomes or some bacterial genomes. The heuristic is not intended for viral genomes or plasmids, either. In addition, similar to binning, circle rescue may produce false circles (Fig. 2) and need to be evaluated with CheckM1.

We also described two reference-free approaches, *k*-mer spectrum and species-level OTUs based on full length 16S rRNAs, to evaluate how well prevalent species in a metagenome sample are represented by a metagenome assembly. Unlike CheckM1, our methods do not depend on known genomes or single-copy core genes and thus are not biased towards existing reference genomes. Applying the methods to real data, we showed that de novo HiFi assembly plus binning can sometimes assemble the great majority of prevalent species into near-complete MAGs with many of them not seen in the existing metagenome catalogs produced from short reads.

## Conclusions

Our work emphasizes the importance of metagenome completeness, which has often been overlooked, by proposing two evaluation methods and examining HiFi metagenome assemblies. We also proposed and implemented a binning method that takes advantage of genome assembly graph of hifiasm-meta, and showed the algorithm could rescue MAGs failed by traditional binners. Nonetheless, we anticipate high-quality metagenome assemblies and further method improvements could transform previously inaccessible approaches, such as analyzing horizontal gene transfers, de novo variant calling in unusual samples, and direct comparison between microbial communities.

## Methods

### Assembly and basic MAG evaluation

We generated HiFi assemblies using hifiasm-meta r73 with default settings for assembly and added "--write-binning" to output fasta files of genome bins. By default, hifiasm-meta writes a tab-delimited table to describe binning result. See Table S2 (Additional file 2) for accession IDs of HiFi datasets [6, 36–43] and Table S4 (Additional file 4) for versions of tools used. Short read assemblies were downloaded from their corresponding studies. Contig coverage for MetaBAT2 and Vamb was estimated with minimap2 alignment and MetaBAT2's jgi module: we ran minimap2 [44] with "`minimap2 -t48 -ak19 -w10 -I10G -g5k -r2k --lj-min-ratio 0.5 -A2 -B5 -O5,56 -E4,1 -z400,50 --sam-hit-only contigs.fa reads.fa.`" BAM file handling used SAMtools [45]. Coverage was estimated by "`jgi_summa_rsize_bam_ contig_depths --outputDepth depth.txt input.bam.`" For binning, we ran MetaBAT2 with "`metab3t2 --seed 1 -i contigs.fa -a depth.txt.`" We ran Vamb with "`vamb -p 48 --outdir ./ --fasta contigs.fa --jgi jgi_depth --minfasta 200000.`" Vamb needs to process inputs in batch, the size of which needs to be lower than the number of input contigs (after length filtering). We use the larger one of either 256 (default) or the round up of length-filtered contig count to the next exponential of two as the batch size. MetaBAT2 or hmBin's random

seed has little influence. For MetaBAT2, Vamb and GraphMb, we separate circular contigs of ≥ 1 Mb into a separate MAG if it is binned together with other contigs, since such tweak helps their performance. We ran GraphMB with "`graphmb –assembly prefix –numcores 48 –num-markers ""` ." We ran SemiBin1 with "`SemiBin single_easy_bin -i asm.fa -b readaln.bam –environment env – sequencing-type long_read` ," where " env" is "human_gut" for libraries prefixed with human-gut, "chicken_caecum" for chicken-gut-1, and "global" for the rest.

We used CheckM1 module "`lineage_wf`" to evaluate MAG quality. Its outputs were formatted by "`checkm qa -o 2`" before parsing. We did not try DAS tools's evaluation in this work, but it should give consistent but more generous results.

We ran rust-mdBG with "`-k 21 -l 14 –density 0.003 -p asm`" then "`magic_simplify_meta  asm`" to generate the final assembly graph and the sequences per developer's recommendation. A freestanding implementation of the circle-finding heuristics was used, and we used mash distance at cutoff 90% to compare between the reported circular paths to merged MAGs of hifiasm-meta's (an earlier version r63, assembly and circle rescue were similar to r73). In env-digester-1, human-gut-3, and sheep-gut-1b, this reported 22, 22, and 120 rescued circles, respectively. We did not try to do MetaBAT2 and bin merging because CheckM1 was sensitive to indels.

### The proposed binning heuristics

Our binning algorithm, hmBin, requires the contig graph and the estimated contig coverage as input. It outputs a FASTA file from the circle finding heuristic and a plain text file describing the binning result.

We apply a depth-first search (DFS) to identify circular paths in the assembly graph. DFS starts with the longest unvisited contig and discovers one circular path at a time. We keep a circular path if it contains ≥ 2 contigs and the path length is ≥ 500 kb. Note a contig may be used in many circular paths that spell similar sequences. To avoid enumerating all these similar paths, at a fork in the graph we prioritize on the contig that has been used less often in previously found circular paths (ties are arbitrarily broken); we also reject a circular path if contigs in the path have been included for ≥ 100 times, in total, in previously found paths. With this threshold, we reduce paths that share many contigs but there may still be redundant circular paths.

We then deduplicate the circles using pairwise Mash distance [18] as follows. If the two circles being compared differ in length for more than 1 Mb, we keep both of them regardless of the Mash distance between them. We only keep the longer circle if the Mash distance between the two circles in comparison is over 95%, a commonly used threshold to determine if two bacterial genomes come from the same species [46, 47]. A threshold of 97% yields similar results when evaluated with CheckM1. Note that the comparison is applied only to circular paths found by the circle-finding heuristic, excluding normally assembled circular contigs. A circle that has low mash distance compared to a long circular contig would not be discarded.

Next, we collect any contig that meets the following criteria as candidate: (1) least 100 kb long, (2) not used in the circle finding heuristics above, and not (3) longer than 1 Mb and circular. We collect two features for each contig, the coverage (read depth) and a canonical 5-mer profile (5-mer frequencies). The coverage is calculated by counting

bases of reads that belong to the contig in question, and all reads that were contained in them. The basepair count is divided by contig length to obtain the coverage. This estimation produced similar results to those from methods based on read alignment (such as "jgi_summarize_bam_contig_depths" of MetaBAT2) in contigs of hundreds of kilobases or longer. The canonical 5-mer profile is simply from *k*-mer counting and normalized by total count. We take the logarithm (base 10) of the coverages then combine the two features to generate a feature matrix of shape (n_samples, 513). The matrix is *z*-score normalized before being embedded to two-dimensional space via Barnes-Hut t-SNE. Hifiasm-meta borrowed Laurens van der Maaten's implementation [48] and the C interface from https://github.com/lh3/bhtsne [49]. To generate bins, we initialize an empty set "block." We examine "seed" contigs starting from the longest candidate, in the embedded space. For each seed, if any other contig that is not already in the "block" is present within a certain radius r1, collect the seed and these neighbors to form a bin. Otherwise, either the seed contig form a bin on its own or at least one circle with radius r1*0.8 exists such that the seed lies on the boundary of the circle and all not-blocked neighbors no farther from the seed than r1*1.6 are covered by the circle. Contigs of a new bin are pushed to "block."

The value of r1 (0.05-0.3) and perplexity of t-SNE (15-50) are insensitive to the results. Using bounding boxes instead of circles produced similar results. 5-mer profile and 4-mer profile (TNF, tetranucleotide frequencies) produced similar results. Partitioning bins t-SNE based on the assembler's knowledge of pairwise contig phasing status made little difference, although this was condition by how hifiasm(-meta) process read overlaps, and we did not try general alignment methods here. Vamb and related works used a version of TNF that accounts for correlations between 4-mers, but our approach was insensitive to this treatment. UMAP produced worse results due to grouping data points into its characteristic trace-like shapes, which resulted in more contaminated bins. We also tried t-SNE as implemented in the python library "sklearn" and R-ct-SNE (Revised Conditional t-SNE) [50] as implemented by its authors which, for our framing of the problem, discourages known non-pairs to be considered neighbors in the embedding of the high dimension space. Read-level phasing information from hifiasm-meta was used as input for R-ct-SNE. R-ct-SNE performed slightly worse in samples with low sequencing coverage, and slightly better in the others, but overall they were similar to each other and the final implementation in hifiasm-meta.

### Examining high read- and assembly-multiplicity *k*-mers

We use human-gut-10 as an example. We identified *k*-mers with at least 2x and up to 15x assembly multiplicity and at least 800x read multiplicity, i.e., the right-most part of *k*-mer spectrum plot above 1x band. Their location on the contigs were collected. We merged overlapping intervals and dumped the sequences. There were 5469 unique sequences (max length 18.6 kb, N50 2.0 kb). We randomly select 20 from these ("seqtk subseq in.fa 20") and did BLAST (blastn web cgi, defaults) against nr/nt. All queries had full length BLAST hits with low sequence divergence and frequently overlap with genes encoding DNA-related enzymes, transposase, and tRNA or rRNA (Additional file 5: Table S5).

### 16S rRNA methods

We identified and annotated 16S rRNA genes from HiFi libraries with the following steps. First, HiFi reads that could align to SILVA reference (specifically, "SILVA_138.1_ SSURef_tax_silva_trunc.fasta") were extracted, with base qualities stripped: "`seqtk subseq hifi.fq<(minimap2 SILVA.fa hifi.fq | cut -f1 | uniq) | seqtk seq -A> SSUreads.fa`." We ran barrnap to identify rRNA genes: "`barrnap --kingdom bac --outseq rRNA.fa SSUreads.fa`." INFERNAL cmsearch might identify a few more rRNAs than barrnap. We believe this would not have major influence on the conclusion based on previous observations. We then annotated rRNA genes with RDP classifier: "`java -Xmx16g -jar RDPTools/classifier.jar classify -o RDP.tsv -h RDP.hier rRNA.fa`." We accept annotations of 16S rRNAs with genus scores of at least 0.9.

To define OTUs from HiFi reads, we first selected 16S genes not marked with "partial" from the barrnap. We used greedy incremental clustering: we initialize an empty collection *S* to collect seed sequences. For each 16S gene *q*, if it could align to any sequence *s* in *S* with alignment block longer than 1000 bp and at least 99% mismatch identity, it is assigned the same OTU label as *s* (if multiple seeds are available, the one with the highest identity will be chosen; if a tie, the seed is arbitrarily chosen from the bests). Otherwise, *q* is added to *S* and assigned a new OTU label. Alignment is done with minimap2's python binding, mappy, with "`preset=map-hifi`." Alignment block length is given by "`mappy.Alignment.blen`." Mismatch identity is calculated as "`mappy.Alignment.mlen/mappy.Alignment.blen`." Assigning OTU label for an unseen 16S copy is done similarly. If a sequence can not align to any seed sequences, its OTU label is undefined.

There are two ways to assign OTU labels to MAGs: (1) collected reads belonging to contigs of a MAG and their OTU labels or (2) identify 16S copies from contigs then assign labels. We did both and found them to be mostly consistent. We only considered near-complete MAGs.

When evaluating MAGs as in Fig. 5 and Fig. S6 (Additional file 7), we drop OTUs with less than 10 16S copies to rule out artifacts from sequencing errors and to ignore species with very low coverage. This was done after the OTU label assignment. A MAG could have more than one OTU assignment. It was difficult to distinguish wrong cases (i.e., suboptimal clustering result) from true cases, i.e., a genome having multiple distinct 16S copies; therefore, we simply accept all OTU labels of a MAG. For example, if the read set yields 3 OTUs (*a*, *b*, and *c*), the assembly has a single MAG from which we identify three 16S copies that are labeled *a*, *a*, *b*. Then, in the plot, both *a* and *b* would be colored as "seen in MAG."

### Supplementary Information

---

**Additional file 1: Table S1.** Supplementary tables and data release used a different sample naming convention. This file provides the name mapping.

**Additional file 2: Table S2.** Sample and HiFi library information.

**Additional file 3: Table S3.** Binning information and evaluation of bins. A tab-delimited table.

**Additional file 4: Table S4.** List of tools their versions used in this manuscript.

---

**Additional file 5: Table S5.** BLAST result summary of 20 high read- and contig- multiplicity *k*-mers intervals.

**Additional file 6: Table S6.** The impact on binner by the post-processing that picks out long circular contigs when they are binned with any other contig. This information is also available from Table 1.

**Additional file 7.** Supplementary figures. **Figure S1.** Comparison of relationships between MAG yield and library size in short read libraries and HiFi libraries. **Figure S2.** Scatter plots showing MAG yields per gigabases (Gb) sequenced. **Figure S3.** Showing all bands of Fig. 5. **Figure S4.** *K*-mer spectrum plots comparing MetaBAT2 merged with rescued circles (mb-merge) and hmBin in all samples. **Figure S5.** *K*-mer spectrum plots using HRGM assemblies as the MAGs, and HiFi reads as the library. **Figure S6.** OTU recovery in samples, with OTU boundary set at 99%, 95% and 90%.

**Additional file 8.** Review history.

### Acknowledgements
Not applicable.

### Peer review information

### Review history
The review history is available as Additional file 8.

### Authors' contributions
HL and XF conceived the projected, carried out the analysis, and wrote the manuscript. All authors read and approved the final manuscript.

### Funding
This work was funded by National Human Genome Research Institute (NHGRI) R01HG010040 and U01HG010961, the Chan Zuckerberg Initiative, and the Sloan Foundation.

### Availability of data and materials
Hifiasm-meta is open source at https://github.com/xfengnefx/hifiasm-meta/ [51]. The yak fork used is at https://github.com/xfengnefx/yam [52]. Scripts and assemblies are at https://github.com/xfengnefx/snpt_mtgncomp and doi:10.5281/zenodo.10868731 [53], respectively. HiFi libraries [6, 40, 41, 43] are all publicly available and can be found under their accession IDs (Additional file 2: Table S2). Short read libraries used in this manuscript can be found in Almeida et al. [21].

## Declarations

**Ethics approval and consent to participate**
Not applicable.

**Consent for publication**
Not applicable.

**Competing interests**
The authors declare that they have no competing interests.

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

## 