## [**Additional file 8.** Review history. · Genome Biology]

Review History

First round of review

Reviewer 1

Are you able to assess all statistics in the manuscript, including the appropriateness of statistical tests used? There are no statistics in the manuscript.

Comments to author:

Feng and Li present here a very interesting approach for the identification of large circular replicons in long-read metagenomes by targeting and extracting long cycles in the assembly graph. I believe this type of approach combining assembly graph exploration with genome-resolved metagenomics is primed to produce an important impact in the field, and continues to be underexplored despite the recent increase in publications exploring this technique. Therefore, I commend the authors on the method presented here and its application to animal gut metagenomes. However, I would like to highlight a number of drawbacks in the manuscript that I consider major issues, as well as a series of minor issues I list below.

Major issues:

1. The presentation of the manuscript is largely anecdotal and rather informal. The authors often use vague or qualitative language when precise data could be presented. These are some examples where this issue emerges:

- P3, L41: "when the assembly was visually reasonably not bad". There is no clear indication of quality metrics or criteria supporting this assertion

- P3, L47: Similarly, despite the existence of formal definitions and extensive literature on the subject, no supporting metrics or criteria are given for "more tangled graphs"

- P5, L57: "a diverse sheep fecal material with extraordinary depth and diversity". No indication is provided on either depth or diversity. The depth could be demonstrated through a k-mer spectrum without normalization (showing the actual accumulation at each redundancy point, not just the relative distribution). On the other hand, several metrics for diversity exist. For example, Nonpareil sequence diversity (Nd) explicitly evaluates sequence diversity and relates that metric to 16S-derived metrics of diversity (<https://doi.org/10.1128/msystems.00039-18>).

- P9, L19: Is there a threshold for "slightly worse but comparable" vs simply "worse"? On the other hand: is there a reason why "worse" is divided in these two categories but "better" isn't? Given the same leeway, how many were "slightly better but comparable"?

- P12, L11: Could you provide a quantitative assessment instead? Which fraction of genes in the 2x band received these annotations? Which fraction of genes in the whole assembly received these annotations? Are the two significantly different?

2. A systematic testing of the method presented here is lacking. The samples used only represent a narrow range of diversity, and all come from very similar environments. Moreover, the key assertions in the manuscript often remain untested. For example, from the title one would expect that the method presented here would significantly increase the fraction of sequence space represented in nearly-complete genomes. However, Figure 3 (and related analyses) only make a direct comparison between rescued circular contigs and the merged collection, but how about the MetaBAT2 MAGs pre-merging? From the experiments presented here, it is impossible to ascertain to what extent the presented method actually increases the representation of sequence space in MAGs with respect to existing methods (such as MetaBAT2). An additional column in Fig. 3 with the kmer spectra for MetaBAT2 MAGs could start to answer that question, and a quantitative evaluation would be even better (see above). Similarly, only MetaBAT2 is tested, but a large number of recent binning techniques has been recently introduced, which could outperform MetaBAT2 for the given samples.

On the other hand, the exceptions presented here are largely dismissed. Perhaps a deeper evaluation could identify the limitations on the technique, guiding future researchers on when to use it. For example, in P10, L41: Any ideas on why these two samples differed? Could Nd (Nonpareil sequence diversity, <https://doi.org/10.1128/msystems.00039-18>) or other metric of diversity possibly have predicted this failure? Similarly, in P12, L41-43 the possibility of miss-assembly, which is the most relevant to this study, is completely ignored.

Finally, no mention is made about the possibility of linear chromosomes, which are prevalent in some bacterial taxa. Similarly, no mention is made of any extra-chromosomal replicons, such as plasmids, which also conform the bacterial genomes.

3. All the comparisons were made against binning of Short Reads (SR) metagenomes. While this choice is partly justified by the authors, I see no good reasons not to apply a traditional binning method to the same LR metagenomes and explicitly test the advantages or disadvantages of the method introduced here against a benchmark that uses the same input data.

In this topic, the experiments presented in P12, L49ff appear to be fundamentally flawed. The authors compare the lengths of the rescued circular contigs against very similar genome assemblies from SR collections. In this comparison, the authors find that circular contigs are larger, and from this conclude that the proposed technique in LR metagenomes produce a more comprehensive representation of the genomes. Note that the rescued circles, by their very nature, are very likely to be complete or nearly-complete (as shown in Fig. 1), and will necessarily be (on average) larger than any reference collection of MAGs filtered by genome identity. In other words: the authors are preselecting complete genomes in their collection, and comparing them against their corresponding MAGs in an unfiltered reference collection. I'm not sure we learnt anything from this comparison.

4. The manuscript requires a thorough reorganization and language check. There are some suggestions below (minor issues), but I would like to highlight the fact that the entire discussion is essentially a mix of methods and results documenting a completely new set of experiments, and should be labeled accordingly. Similar, the results in the Results section have extensive discussion interspersed, but no cohesive discussion section is developed.

5. The code appears not to be ready for production. Indeed, the authors don't claim to provide ready-to-use Software, but I question this decision, as any potential readers would be interested primarily to test this method in their own LR samples. At the very least, I would urge the authors to document the use of the main scripts in their repository.

Minor issues:

1. The title is simultaneously misleading and uninformative. I consider this is a rather minor issue, but note that the title does not tell us what the method (or the manuscript) is about. For example, "bacterial contents" is assumed to refer to "genomic data", given the pairing with "metagenomic samples", but there appear to be no reason or need for this implicit assumption. Also, note that the tests presented here don't actually demonstrate that this method advances "towards complete representation". Finally, the method is only tested in a limited collection with very narrow range of diversity, so it would be appropriate even from the title to state that this method only works for gut microbiome (or demonstrate otherwise).

2. P1, L48: It should probably be "elusive"

3. P1, L56: How do you define "genome-complete" here? If they're actually complete, there should be no assembly gaps (other than those between replicons)

4. P1, L61: This is unfortunately true, but it would be worth highlighting the counterexamples in the literature. For example, see <https://doi.org/10.1111/1462-2920.15112>, in which this is explicitly evaluated.

5. P2, L6: I'm not sure what the authors mean by "approve" here, please reword for clarity

6. P2, L10: Very rarely are 16S sequences actually used as proxy for species definition. In the first cited reference, 16S identity is used as a lower-boundary above which additional information is necessary. In the other two references the definitions are operational (OTUs), not necessarily for the species level.

7. P2, L15: Remove "a" or make "catalogs" singular

8. P2, L19: What do you mean by "their quality MAGs"?

9. P2, L31: This entire sentence appears to be a non sequitur

10. P2, L34: Note that the paper referenced above (<https://doi.org/10.1111/1462-2920.15112>) explicitly tests this point and provides evidence towards the point made by the authors on the issue of composition inference. The two references cited here, in contrast, use genomic profiling but don't demonstrate any clear issues with "composition inference".

11. P2, L41: Should be "problematic"

12. P2, L41-44: I could not understand this sentence, please rephrase for clarity

13. P2, L45-48: Please provide either a reference or testing data for this assertion

14. P2, L51: Should be "learning"
15. P2, L53-55: Please rephrase for clarity
16. P3, L29: Should be "... do not perform as well in..."
17. P3, L31: Please rephrase for precision. I would suggest making an explicit distinction between "completeness" (estimand) and "estimated completeness" (estimator). It is unclear to me if the authors are indicating an issue with the completeness estimation (i.e., that the estimation is spatially biased and therefore can be inflated due to the accumulation of marker genes in a long contig), or that a large fraction of the genome assembly is represented by a single contig. If the latter, "completeness" doesn't really have a place here.
18. P3, L47: Please provide either a reference or testing data for this assertion
19. Fig. 1: The green band should be " $\geq 1\text{Mb}$ ", not " $> 1\text{Mb}$ "
20. P5, L38 and elsewhere: Please revise the use of "circles" vs "cycles". If they are meant to be different entities in the context of the manuscript, please define them explicitly. Otherwise, I would recommend using only one of the two consistently to prevent unnecessary confusion.
21. P5, L46: Should be "quality"
22. P5, L58: Should be "heuristics"
23. P8, L12: No indication on what this command-line fragment means or to which software does it apply
24. Figure 4: I believe "poorly and good" should be "good and poorly"
25. P9, L31: What "convenience" does this padding offer? I would imagine any software used here supports multi-fasta format, so this padding appears entirely unnecessary (and potentially counterproductive, as recognized in L32).
26. P9, L59: Should be "... presented in tools such as KAT..."
27. P15, L46: Should be "a metagenome"

Reviewer 2

Are you able to assess all statistics in the manuscript, including the appropriateness of statistical tests used? No, I do not feel adequately qualified to assess the statistics.

Comments to author:

The manuscript "Towards complete representation of bacterial contents in metagenomic samples" by Feng and Li describes a new addition to hifiasm-meta that enable recovery of

dominant genomes under strain-diversity. Furthermore, they describe a visualization using kmers to analyse what abundance-graded fraction of the metagenome that have assembled.

I found the overall approaches highly interesting, both the genome binning in light of strain diversity and the abundance based visualization of recovery rate of MAGs. However, while I can understand the major points of the manuscript, the language is often imprecise, which hinders comprehension and might make some refrain from reading the manuscript. I feel that this is unfortunate as the findings in the paper are interesting. Combined with a lack of references to previous relevant work and comparison with state-of-the-art bidders i've not conducted an in depth review, but instead provided some general guidelines for what needs to be improved before a proper review can be conducted.

Below I've given a few examples from the first pages of the manuscript, but the paper needs a complete and careful rewrite.

p1-120-21: "we may think about." The phrase is an example of "talking" language, and this needs to be rephrased throughout the paper.

p1-154-60: "Most de novogenome-complete [4, 5] MAGs still contain an average of 87 assembly gaps with median length about 1.3kb. We manually checked some and found that these gaps either have no presence in the BLAST nr/nt database using BLASTn, or were homologous to shared genes such as ribosomal RNA (rRNA) operons." The sentence is very hard to understand in its current form. I assume you mean: "Most HQ-MAGs generated using short read sequencing still contain 50+ contigs. This is due to repeat elements longer than the sequencing read length, often transposons or ribosomal rRNA genes (ref)." This is relatively common knowledge that repeats (i.e., identical sequences > read length) breaks assemblies. This is true both for pure culture sequencing and metagenomics. A good and almost ten year old reference I usually use is this: <https://doi.org/10.1016/j.mib.2014.11.014>

p1-160-63: "Second, MAGs are rarely checked for their representation-completeness. Studies often assumed that sufficiently abundant or the most abundant species will be reconstructed [6, 7, 8]." The sentence is very hard to understand in its current form. I assume you mean: "It is rarely investigated if the recovered MAGs are representative of the entire sequenced population". I do not think this is true; in many studies, they calculate how large a fraction of the total bases in your metagenome that your MAGs cover. Again, I think it is relatively common knowledge that strain-diversity limits assembly and MAG recovery. There are numerous papers describing this or developing tools to do strain-resolved metagenomics by looking at SNP rates etc etc. Just to name one, you can look at inStrain with 100's of citations <https://www.nature.com/articles/s41587-020-00797-0>.

p2-19-10: "One major obstacle for improving the situation was that 16S rRNA sequences, which is a proxy of species definition". This makes no sense.

Other:

Using the assembly graph to improve binning has been done for many years. The first tools used paired-end connections as a proxy for the assembly graph (e.g., CONCOCT, <https://www.nature.com/articles/nmeth.3103>), but since several tools have emerged that use the assembly graph directly (see e.g., GraphMB and references herein <https://academic.oup.com/bioinformatics/article/38/19/4481/6668279>). Furthermore, there are long-read optimized bidders available, see e.g., SemiBin (<https://www.nature.com/articles/s41467-022-29843-y>, the newest release is updated to handle long-read assemblies) and GraphMB (<https://academic.oup.com/bioinformatics/article/38/19/4481/6668279>). In order to provide state-of-the-art comparisons, it should be done against state-of-the-art bidders that take advantage of long read assemblies e.g., SemiBin and GraphMB. The comparison against MetaBat2 is still relevant as it represents the most widely used binning software.

p3-151: The assumption of genome sizes in the range of 1-8 Mbp seems too narrow. There are numerous examples of genomes below 1 Mbp (a large part of the candidate phyla). A more reasonable range would be 0.5 Mbp to 15 Mbp (from memory, the largest bacterial genome is approx 17 Mbp).

The kmer-spectrum plots are nice and a great way to visualize what is not assembled/binning. However, kmer counting has had numerous applications, and I'm not sure if the kmer-spectrum plots are novel? See these two relatively old articles on the use of kmer's in analysis and QC and look up related references (<https://journals.plos.org/plosone/article?id=10.1371/journal.pone.0101271> and <https://academic.oup.com/bioinformatics/article/33/4/574/2664339>)

Although still a preprint, it would also be useful to compare with the "stRainy approach" from the Flye developers and also be inspired of their use of references <https://www.biorxiv.org/content/10.1101/2023.01.31.526521v1>

Summary of responses:

We thank both reviewers for the thorough review and the detailed comments. We have rewritten over half of the manuscript, included five more HiFi datasets and evaluated two more binning algorithms (GraphMB and SemiBin). We have also fixed bugs and performance issues with hifiasm-meta, improved the circle finding heuristic and implemented an integrated binning algorithm in hifiasm-meta. Point-by-point responses are shown below:

Reviewer #1

Feng and Li present here a very interesting approach for the identification of large circular replicons in long-read metagenomes by targeting and extracting long cycles in the assembly graph. I believe this type of approach combining assembly graph exploration with genome-resolved metagenomics is primed to produce an important impact in the field, and continues to be underexplored despite the recent increase in publications exploring this technique. Therefore, I commend the authors on the method presented here and its application to animal gut metagenomes. However, I would like to highlight a number of drawbacks in the manuscript that I consider major issues, as well as a series of minor issues I list below.

Q1.1.1: The presentation of the manuscript is largely anecdotal and rather informal. The authors often use vague or qualitative language when precise data could be presented. These are some examples where this issue emerges:

- P3, L41: "when the assembly was visually reasonably not bad". There is no clear indication of quality metrics or criteria supporting this assertion
- P3, L47: Similarly, despite the existence of formal definitions and extensive literature on the subject, no supporting metrics or criteria are given for "more tangled graphs"
- P5, L57: "a diverse sheep fecal material with extraordinary depth and diversity". No indication is provided on either depth or diversity. The depth could be demonstrated through a k-mer spectrum without normalization (showing the actual accumulation at each redundancy point, not just the relative distribution). On the other hand, several metrics for diversity exist. For example, Nonpareil sequence diversity (Nd) explicitly evaluates sequence diversity and relates that metric to 16S-derived metrics of diversity (<https://journals.asm.org/doi/10.1128/mSystems.00039-18>).
- P9, L19: Is there a threshold for "slightly worse but comparable" vs simply "worse"? On the other hand: is there a reason why "worse" is divided in these two categories but "better" isn't? Given the same leeway, how many were "slightly better but comparable"?
- P12, L11: Could you provide a quantitative assessment instead? Which fraction of genes in the 2x band received these annotations? Which fraction of genes in the whole assembly received these annotations? Are the two significantly different?

A1.1.1: We thank the reviewers for the suggestions. We no longer mention these quoted sentences in the revised manuscript. In particular, we have added a new Figure 1 to show local assembly subgraphs. This gives the context of our previous wording in P3-L41 and P3-L47. On P5-L57, we now provide the Nd metric in Table 1. On P9-19, we give precise numbers as follows:

"Across all tested datasets, 74 near-complete MetaBAT2 bins were discarded by the procedure above. 64 of them have counterparts in the collection of the rescued circles with <1% mash distance (52 of them <0.1%). Out of the 64 pairs, 61 rescued circles matches were better than the rejected MetaBAT2 bins in terms of equal or better CheckM completeness and contamination. The rest three pairs are: 1) rejected MetaBAT2 bin had 100.0% completeness

and 0.0% contamination, its best rescued circle counterpart had 98.8% and 0.48%, 2) rejected bin had 99.33% completeness and 1.34% contamination, its counterpart 97.32% and 1.34%, and 3) rejected bin had 90.7% completeness and 0.94% counterpart, its counterpart 99.06% and 2.96%.”

On P12-L11, we have changed the sentence to: “*We manually examined a few k-mers with their flanking region, and found them to harbor ubiquitous genes (e.g. tRNAs) or horizontally transferable sequences*”. This is only a side point; a systematic analysis would require detailed gene annotation of all MAGs.

Q1.1.2: A systematic testing of the method presented here is lacking. The samples used only represent a narrow range of diversity, and all come from very similar environments.

A1.1.2: We would love to try our methods on a variety of samples. However, at the time we wrote the initial manuscript, we could only find a few non-gut HiFi samples in SRA. In this revision, we added the following samples/libraries (for base count, QV and read N50, see Table S4):

- ERR10905741: UK Industrial Anaerobic Digesters
- ERR10905742: UK Industrial Anaerobic Digesters
- ERR10905743: UK Industrial Anaerobic Digesters
- ERR4811462: Unspecified, but likely sludge
- DRR290133: Hot spring sediment microbiome

The first four datasets were available in 2023, months after our initial submission. The last one was made public a month before our submission.

Q1.1.3: Moreover, the key assertions in the manuscript often remain untested. For example, from the title one would expect that the method presented here would significantly increase the fraction of sequence space represented in nearly-complete genomes. However, Figure 3 (and related analyses) only make a direct comparison between rescued circular contigs and the merged collection, but how about the MetaBAT2 MAGs pre-merging? From the experiments presented here, it is impossible to ascertain to what extent the presented method actually increases the representation of sequence space in MAGs with respect to existing methods (such as MetaBAT2). An additional column in Fig. 3 with the kmer spectra for MetaBAT2 MAGs could start to answer that question, and a quantitative evaluation would be even better (see above).

A1.1.3: We have changed the title to “*Evaluating and improving the representation of bacterial contents in metagenomic samples*”. We have added a new Figure 4 to show how our post-processing step and circle rescue heuristic improve existing binning algorithms on each dataset.

Q1.1.4: Similarly, only MetaBAT2 is tested, but a large number of recent binning techniques has been recently introduced, which could outperform MetaBAT2 for the given samples.

A1.1.4: In the original manuscript, we evaluated both MetaBAT2 and vamb as standalone binners, though we only tried to combine our circle rescue algorithm with MetaBAT2. In the revised manuscript, we additionally evaluated graphMB and SemiBin and combined circle rescue with all evaluated binning algorithms. On our datasets, MetaBAT2 is consistently better than vamb and graphMB, and is only second to SemiBin. It is worth noting that SemiBin makes

extensive use of single copy marker genes, the main signal checkM v1 uses for evaluation. As a result, SemiBin has the same biases and limitations as checkM. It is actually unfair to use checkM to compare SemiBin to other bidders but we still include SemiBin for a reference.

Q1.1.5: On the other hand, the exceptions presented here are largely dismissed. Perhaps a deeper evaluation could identify the limitations on the technique, guiding future researchers on when to use it. For example, in P10, L41: Any ideas on why these two samples differed? Could Nd (Nonpareil sequence diversity) or other metric of diversity possibly have predicted this failure? Similarly, in P12, L41-43 the possibility of miss-assembly, which is the most relevant to this study, is completely ignored.

A1.1.5: As our algorithm is topology based, its effectiveness is mainly determined by the topology of the underlying assembly graph. If there are many circle-like subgraphs or regions, our algorithm can usually separate them out and improve the assembly's completeness. However, if the assembly graph only contains simple circular contigs or visually there are no circular structures in the assembly graph (see the screenshot below), our algorithm will not improve the assembly. We do not have a good quantitative predictor for the success of binning. In the revision, we provide the Nd metric Nonpareil3 reports for each dataset. There is not a strong correlation between Nd and binning results, either.

On P12, L41-43 in the original manuscript, we mentioned “*collapsing of very similar haplotypes during assembly*”. These are not misassemblies as naturally there are possibly many haplotypes in a metagenome sample. In fact, hifiasm-meta is more powerful than metaFlye on separating similar haplotypes, much more so than short-read assemblers. We have still removed this sentence to avoid confusion.

^ A sample (sludge) that did not assemble well. A lot of contigs are both short and have high degrees. The circle finding heuristics could work if the assembly step was able to generate at least some long contigs in these regions, but otherwise will not.

Q1.1.6: Finally, no mention is made about the possibility of linear chromosomes, which are prevalent in some bacterial taxa. Similarly, no mention is made of any extrachromosomal replicons, such as plasmids, which also conform the bacterial genomes.

A1.1.6: Linear chromosomes are not rescued due to the nature of our algorithm. We have added a new paragraph in Discussions that mentioned this weakness: “*Because the circle rescue heuristic only looks for circular paths, it will naturally miss linear chromosomes in small Eukaryotic genomes or some bacterial genomes*”.

The quality of plasmids and phages is hard to evaluate. When we wrote our first hifiasm-meta paper, we tried PlasmidVerify and ViralVerify to evaluate the completeness of plasmids and phages (see Table S5 and S6 in PMID:35534630). However, the results were sensitive to hyperparameters, e.g. sequence similarity cutoff, scores provided by the tools etc, and the total number of plasmids or viruses could differ for more than ten folds between methods. We thus did not evaluate them in this work. In the revision, we now acknowledge that “*The [circle rescue] heuristic is not intended for viral genomes or plasmids, either*”.

Q1.1.7: All the comparisons were made against the binning of Short Reads (SR) metagenomes.

While this choice is partly justified by the authors, I see no good reasons not to apply a traditional binning method to the same LR metagenomes and explicitly test the advantages or disadvantages of the method introduced here against a benchmark that uses the same input data.

A1.1.7: We would like to clarify that we compared binned LR metagenomes against binned SR metagenomes to demonstrate LR bins have less gaps and therefore tend to be larger than their SR counterparts. We have moved the related figures to Supplementary materials. With the addition of standalone binning algorithms (**A1.1.4**), we now explicitly compare our method to traditional binning methods in this revision and show we gain more near-complete MAGs with our post-binning-processing and the circle rescue heuristic (**A1.1.3**).

Q1.1.8: In this topic, the experiments presented in P12, L49ff appear to be fundamentally flawed. The authors compare the lengths of the rescued circular contigs against very similar genome assemblies from SR collections. In this comparison, the authors find that circular contigs are larger, and from this conclude that the proposed technique in LR metagenomes produce a more comprehensive representation of the genomes. Note that the rescued circles, by their very nature, are very likely to be complete or nearly-complete (as shown in Fig. 1), and will necessarily be (on average) larger than any reference collection of MAGs filtered by genome identity. In other words: the authors are preselecting complete genomes in their collection, and comparing them against their corresponding MAGs in an unfiltered reference collection. I'm not sure we learnt anything from this comparison.

A1.1.8: We agree with the reviewer that HiFi MAGs should be longer more often than not, therefore the lines and its figure in question is not informative. In the revised manuscript, we only mention the MAG size comparisons with SR MAGs as a sanity check.

Q1.1.9: The manuscript requires a thorough reorganization and language check. There are some suggestions below (minor issues), but I would like to highlight the fact that the entire discussion is essentially a mix of methods and results documenting a completely new set of experiments, and should be labeled accordingly. Similarly, the results in the Results section have extensive discussion interspersed, but no cohesive discussion section is developed.

Q1.1.9: We have rewritten more than half of the manuscript. Point to point answers to the editing suggestions are listed in **A1.1.12**.

Q1.1.10: The code appears not to be ready for production. Indeed, the authors don't claim to provide ready-to-use Software, but I question this decision, as any potential readers would be interested primarily to test this method in their own LR samples. At the very least, I would urge the authors to document the use of the main scripts in their repository.

A1.1.10: The latest hifiasm-meta comes with integrated binning and circle rescue. These are now implemented in C++ and compiled into the hifiasm-meta executable. We additionally provide a standalone Python script for circle rescue and a script for merging rescued circles with third-party binners. They are available at github repo xfengnefx/snpt_mtgncomp. We have added more documentations for the two scripts.

Reviewer#1 Minor:

Q1.1.11: The title is simultaneously misleading and uninformative. I consider this is a rather minor issue, but note that the title does not tell us what the method (or the manuscript) is about.

For example, "bacterial contents" is assumed to refer to "genomic data", given the pairing with "metagenomic samples", but there appear to be no reason or need for this implicit assumption. Also, note that the tests presented here don't actually demonstrate that this method advances "towards complete representation". Finally, the method is only tested in a limited collection with very narrow range of diversity, so it would be appropriate even from the title to state that this method only works for gut microbiome (or demonstrate otherwise).

A1.1.11: We have changed the title to "*Evaluating and improving the representation of bacterial contents in long-read metagenome assemblies*" (see also **A1.1.3**). We have also added more recent data (see also **A1.1.2**).

Q1.1.12:

2. P1, L48: It should probably be "elusive"
3. P1, L56: How do you define "genome-complete" here? If they're actually complete, there should be no assembly gaps (other than those between replicons)
4. P1, L61: This is unfortunately true, but it would be worth highlighting the counterexamples in the literature. For example, see <https://ami-journals.onlinelibrary.wiley.com/doi/10.1111/1462-2920.15112> , in which this is explicitly evaluated.
5. P2, L6: I'm not sure what the authors mean by "approve" here, please reword for clarity
6. P2, L10: Very rarely are 16S sequences actually used as proxy for species definition. In the first cited reference, 16S identity is used as a lower-boundary above which additional information is necessary. In the other two references the definitions are operational (OTUs), not necessarily for the species level.
7. P2, L15: Remove "a" or make "catalogs" singular
8. P2, L19: What do you mean by "their quality MAGs"?
9. P2, L31: This entire sentence appears to be a non sequitur
10. P2, L34: Note that the paper referenced above explicitly tests this point and provides evidence towards the point made by the authors on the issue of composition inference. The two references cited here, in contrast, use genomic profiling but don't demonstrate any clear issues with "composition inference".
11. P2, L41: Should be "problematic"
12. P2, L41-44: I could not understand this sentence, please rephrase for clarity
13. P2, L45-48: Please provide either a reference or testing data for this assertion
14. P2, L51: Should be "learning"
15. P2, L53-55: Please rephrase for clarity
16. P3, L29: Should be "... do not perform as well in..."
17. P3, L31: Please rephrase for precision. I would suggest making an explicit distinction between "completeness" (estimand) and "estimated completeness" (estimator). It is unclear to me if the authors are indicating an issue with the completeness estimation (i.e., that the estimation is spatially biased and therefore can be inflated due to the accumulation of marker genes in a long contig), or that a large fraction of the genome assembly is represented by a single contig. If the latter, "completeness" doesn't really have a place here.
18. P3, L47: Please provide either a reference or testing data for this assertion
19. Fig. 1: The green band should be "≥1Mb", not ">1Mb"
20. P5, L38 and elsewhere: Please revise the use of "circles" vs "cycles". If they are meant to be different entities in the context of the manuscript, please define them explicitly. Otherwise, I would recommend using only one of the two consistently to prevent unnecessary confusion.
21. P5, L46: Should be "quality"
22. P5, L58: Should be "heuristics"

- 23. P8, L12: No indication on what this command-line fragment means or to which software does it apply
- 24. Figure 4: I believe "poorly and good" should be "good and poorly"
- 25. P9, L31: What "convenience" does this padding offer? I would imagine any software used here supports multi-fasta format, so this padding appears entirely unnecessary (and potentially counterproductive, as recognized in L32).
- 26. P9, L59: Should be "... presented in tools such as KAT..."
- 27. P15, L46: Should be "a metagenome"

A1.1.12: We are grateful to the reviewer for the thorough comments. As we have largely rewritten the first 9 pages in the initial submission, most of these issues have been addressed. On a few remaining questions:

18. We are attaching the metaFlye assembly graph for the env-digester-2 dataset. The metaFlye graph has a large and complex subgraph on the top, while the hifiasm-meta graph is simpler. Furthermore, the metaFlye assembly graph has different characteristics that would cripple our circle finding algorithm. Our revised manuscript provides details: *"MetaFlye does not produce an assembly graph for contigs. It instead generates a repeat graph which, unlike string graphs used by hifiasm-meta, may reuse a unitig in multiple contigs. Our circle finding algorithm only traverses each unitig or contig once and does not work with metaFlye assembly graphs. In theory, it is possible to adapt our algorithm for repeat graphs. However, metaFlye assembly graphs more often look like the one in Figure 1B. We would not be able to get good results anyway."*

25. The revised version no longer uses paddings when evaluating the bins.

Reviewer #2

The manuscript "Towards complete representation of bacterial contents in metagenomic samples" by Feng and Li describes a new addition to hifiasm-meta that enable recovery of dominant genomes under strain-diversity. Furthermore, they describe a visualization using kmers to analyse what abundance-graded fraction of the metagenome that have assembled.

I found the overall approaches highly interesting, both the genome binning in light of strain diversity and the abundance based visualization of recovery rate of MAGs. However, while I can understand the major points of the manuscript, the language is often imprecise, which hinders comprehension and might make some refrain from reading the manuscript. I feel that this is unfortunate as the findings in the paper are interesting. Combined with a lack of references to previous relevant work and comparison with state-of-the-art bidders i've not conducted an in depth review, but instead provided some general guidelines for what needs to be improved before a proper review can be conducted.

Q1.2.1: Below I've given a few examples from the first pages of the manuscript, but the paper needs a complete and careful rewrite.

p1-l20-21: "we may think about." The phrase is an example of "talking" language, and this needs to be rephrased throughout the paper.

p1-l54-60: "Most de novo genome-complete [4, 5] MAGs still contain an average of 87 assembly gaps with median length about 1.3kb. We manually checked some and found that these gaps either have no presence in the BLAST nr/nt database using BLASTn, or were homologous to shared genes such as ribosomal RNA (rRNA) operons." The sentence is very hard to understand in its current form. I assume you mean: "Most HQ-MAGs generated using short read sequencing still contain 50+ contigs. This is due to repeat elements longer than the sequencing read length, often transposons or ribosomal rRNA genes (ref)." This is relatively common knowledge that repeats (i.e., identical sequences > read length) breaks assemblies. This is true both for pure culture sequencing and metagenomics. A good and almost ten year old reference I usually use is this:
<https://www.sciencedirect.com/science/article/pii/S1369527414001817?via%3Dihub>

Q1.2.1: We thank the reviewer for the suggestion. We have rewritten half of the main text and tried to reduce such phrases. We do not mention the quoted sentence on p1-l54-60, either. For clarification, we had run BLAST in the initial submission to check whether there are genes in short-read assembly gaps.

Q1.2.2: p1-l60-63: "Second, MAGs are rarely checked for their representation-completeness. Studies often assumed that sufficiently abundant or the most abundant species will be reconstructed [6, 7, 8]." The sentence is very hard to understand in its current form. I assume you mean: "It is rarely investigated if the recovered MAGs are representative of the entire sequenced population".

I do not think this is true; in many studies, they calculate how large a fraction of the total bases in your metagenome that your MAGs cover. Again, I think it is relatively common knowledge that strain-diversity limits assembly and MAG recovery. There are numerous papers describing this

or developing tools to do strain-resolved metagenomics by looking at SNP rates etc etc. Just to name one, you can look at inStrain with 100's of citations.

A1.2.2: We have rewritten the Background section and removed the quoted sentence. We have acknowledged that some papers do map reads back to MAGs to check completeness. Nonetheless, this simple strategy is limited. In the revised manuscript, we said "*We could map reads back to the assembly and check how many reads are unmapped. This unfortunately does not tell us the characteristics of under-represented species and thus does not inform how to improve our assemblies*". Our k-mer and 16S RNA based evaluation tell us whether abundant species are missing and what species are missing based on 16S.

Also for clarification, when we wrote the hifiasm-meta paper, we paid particular attention to strain diversity. Note that long-read and short-read assembly algorithms behave distinctly. For example, hifiasm-meta can sometimes completely separate strains at a few percent sequence divergence (see also Figure 3). Short-read assembly would collapse such strains together. It is not straightforward to transfer knowledge learned on short-read assembly to long-read assembly.

Q1.2.3: p2-I9-10: "One major obstacle for improving the situation was that 16S rRNA sequences, which is a proxy of species definition". This makes no sense.

A1.2.3: We have rewritten the Background section and removed this sentence.

Q1.2.4: Using the assembly graph to improve binning has been done for many years. The first tools used paired-end connections as a proxy for the assembly graph (e.g., CONCOCT), but since several tools have emerged that use the assembly graph directly (see e.g., [GraphMB](https://academic.oup.com/bioinformatics/article/38/19/4481/6668279) and references herein <https://academic.oup.com/bioinformatics/article/38/19/4481/6668279>). Furthermore, there are long-read optimized binners available, see e.g., [SemiBin](https://www.nature.com/articles/s41467-022-29843-y) (<https://www.nature.com/articles/s41467-022-29843-y>, the newest release is updated to handle long-read assemblies) and [GraphMB](https://academic.oup.com/bioinformatics/article/38/19/4481/6668279) (<https://academic.oup.com/bioinformatics/article/38/19/4481/6668279>). In order to provide state-of-the-art comparisons, it should be done against state-of-the-art binners that take advantage of long read assemblies e.g., [SemiBin](https://academic.oup.com/bioinformatics/article/38/19/4481/6668279) and [GraphMB](https://academic.oup.com/bioinformatics/article/38/19/4481/6668279). The comparison against [MetaBat2](https://academic.oup.com/bioinformatics/article/38/19/4481/6668279) is still relevant as it represents the most widely used binning software.

A1.2.4: We thank the reviewer for the suggestion. In the revision, we have evaluated [GraphMD](https://academic.oup.com/bioinformatics/article/38/19/4481/6668279) and [SemiBin](https://academic.oup.com/bioinformatics/article/38/19/4481/6668279) in addition to [MetaBAT2](https://academic.oup.com/bioinformatics/article/38/19/4481/6668279) and [vamb](https://academic.oup.com/bioinformatics/article/38/19/4481/6668279). We also added a reference-free and alignment-free binning step to hifiasm-meta. Please see answer [A1.1.4](https://academic.oup.com/bioinformatics/article/38/19/4481/6668279) to reviewer 1 for details.

Q1.2.5: p3-I51: The assumption of genome sizes in the range of 1-8 Mbp seems too narrow. There are numerous examples of genomes below 1 Mbp (a large part of the candidate phyla). A more reasonable range would be 0.5 Mbp to 15 Mbp (from memory, the largest bacterial genome is approx 17 Mbp).

A1.2.5: We have relaxed the 1Mbp threshold to 500Kbp and removed the 8Mbp restriction. The main conclusion remains the same.

Q1.2.6: The kmer-spectrum plots are nice and a great way to visualize what is not assembled/binning. However, kmer counting has had numerous applications, and I'm not sure if the kmer-spectrum plots are novel? See these two relatively old articles on the use of kmer's in analysis and QC and look up related references (ref1:

<https://journals.plos.org/plosone/article?id=10.1371/journal.pone.0101271> , ref2:
<https://academic.oup.com/bioinformatics/article/33/4/574/2664339>).

A1.2.6: Our kmer-spectrum plot is indeed inspired by KAT (ref2). In our manuscript, we acknowledged that “*we borrowed and adapted the k-mer spectrum plot presented in tools such as KAT and Merqury*”. Nonetheless, designed for single-sample assembly, the original KAT and Merqury plots would not be informative for metagenomic samples with highly variable k-mer content. This is why we developed our own version. Our plots are also different from the metagenomic applications described in ref1. To the best of our knowledge, no other papers have used plots similar to ours.

Q1.2.7: Although still a preprint, it would also be useful to compare with the “stRainy approach” from the Flye developers and also be inspired of their use of references (url).

A1.2.7: The stRainy developers did not apply the tool to hifiasm-meta assembly and without stRainy, hifiasm-meta already performed the best on the HiFi datasets in their paper. This suggests stRainy was mainly developed for metaFlye that could not resolve very similar strains. It may not improve hifiasm-meta further.

We tried stRainy anyway. We ran stRainy (commit aaafde) with minimap2 2.24-r1122 on hifiasm-meta’s primary assemblies of all samples used in the revised manuscript, but no run finished successfully. Some contigs would fail at the Flye polisher step with the error message “Error running the Flye polisher. Make sure the fasta file contains only the primary alignments”. How to fix the issue was not apparent to us.

Command use for one sample:

```
grep -v '^A' asm.p_ctg.gfa >asm.noA.p_ctg.gfa # formatting
```

```
python stRainy/strainy.py -t 64 -g asm.noA.p_ctg.gfa -q hifi_reads.fq.gz -o output -m hifi
```

We note that metaflye produces shorter contigs than hifiasm-meta, and stRainy might not be able to make up the gap. For example, in human-gut-3, metaflye v2.9.2 produced 134 \geq 500kb contigs, 32 of them near-complete, 6 high-quality and 10 medium quality. Hifiasm-meta produced 123, but 49 of them were near-complete, 1 high-quality and 8 medium-quality. For env-digestor-2, metaflye produced 127 \geq 500kb contigs, 19/15/11 for the three quality brackets. Hifiasm-meta produced 212 \geq 500kb contigs and 38/27/19.

Second round of review

Reviewer 1

I commend the authors on a much-improved version of their manuscript. I thoroughly enjoyed this reading. I also appreciate the detailed responses in the author's rebuttal, which I found convincing and well supported. I have only minor concerns with respect to the current version of the manuscript:

Minor issues

1. In P4, L27, I believe the authors meant to write that "512 is the total number of canonical 5-mers". Of course, 512 is also the number of non-canonical 5-mers, since 5 is an odd number and 5-mers cannot be identical to their reverse-complement, so the phrase is technically correct. However, I see no reason to confuse the reader with this.
2. In P9, L16 and throughout the manuscript the authors have left references to "Figure ??". Is there any information missing from the submission?
3. In P9, L21, the authors write "within genome-wide read coverage range". It wasn't clear to me what this meant. Does this refer to the distribution of sequencing breadths of all genomes in the metagenome? Please clarify.
4. The discussion of the manuscript retains a large portion of previously unreported results (P12, L36 to P13, L30). This is a matter of preference, but I believe it would be clearer for readers if this fragment of text was transferred to its own sub-section in Results, and only briefly discussed in the Discussion section.
5. In P13, L40ff, the authors write "..., bringing MAGs closer to their original goal which is to delegate their biosample with minimum bias". I disagree with this statement. The earliest attempts at genome-resolved metagenomics intended to retrieve genomes from some populations, but not to do community ecology from these genome catalogues. Indeed, even today, the large majority of studies using MAGs address either aspects of population ecology or other species-specific aspects such as microdiversity, metabolic potential, distribution, etc. This could be an aim of some studies (and a very worthy aim!) but I don't think this was the "original goal".
6. In P14, L62 and elsewhere in the manuscript, the authors use the phrase "at least roughly 500Kb" or similar. Is there a reason why an exact number cannot be given instead? Is the algorithm adaptively changing that threshold? If so, how is this done? Or, if the authors mean that it's an implementation parameter that can be changed, with a default value of 500Kb, I would urge the authors to indicate this explicitly instead.
7. In P15, L6, the authors write "contains a non-significant amount of contigs". What is that amount? Is it a specific fraction of the contigs in the DFS? Is it a set number of contigs? How is this "significant amount" determined?
8. In P15, L10ff the authors explain when contigs are double-counted, but it wasn't clear to me what the "and so on so forth" is meant to indicate here. How is the pattern continued?

Additionally, I would recommend the authors revise the grammar of the manuscript thoroughly. I am convinced that this does not deter from the high scientific quality of the paper, but addressing some issues could facilitate the reading of the manuscript. During my own reading I collected a few suggestions, but this is not to be considered a comprehensive language revision:

1. P2, L28: "as whole" should be "as a whole"

2. P2, L33: “tempted” should be “tempting”
3. P2, L47: “abundance” should be “abundant”
4. P3, L48: “... (DFS) on the assembly...” should be “... (DFS) traversals on the assembly...”
5. P3, L52: “removes” should be “remove”
6. P6, L47: “by” should be “but”
7. P7, L14: “rest” should be “remaining”
8. P9, L12: “resulted” should be “resulting”
9. P15, L34: “such” should be “it”
10. P15, L54: “staring” should be “starting”

Reviewer 2

The manuscript "Evaluating and improving the representation of bacterial contents in long-read metagenome assemblies" by Feng and Li describes a new addition to hifiasm-meta that enable recovery of dominant genomes under strain-diversity. Furthermore, they describe a visualization using kmers to analyse what abundance-graded fraction of the metagenome that have assembled. I found the overall approaches highly interesting, both the genome binning in light of strain diversity and the abundance based visualization of recovery rate of MAGs. I previously reviewed the manuscript and it is now much easier to understand what the authors are trying to do.

Major comment 1: How to resolve strains

I would like a more in depth introduction, results and discussion on the "rescue circle" part of the paper. To me, this is where most of the scientific contribution of this paper lies. As I understand, you go through the assembly graph and try to recover circular genomes. In the presence of strain diversity or extensive HGT (not that large a problem in long-read data) the assembly graph forks, as nicely showed in your figure 1A.

In the simplest case you have 2 strains and the forks then represents SNP's or entire genes that are different between the two strains. What is very unclear to me is how you take one or another path in the graph. It is stated that you do a "Depth-first search (DFS)" but you do not describe what that is? Do you always take the path with the highest coverage through the graph? And how do you handle examples with strains in similar abundance, or when many strains exist? I'm a little concerned that the algorithm just put contigs together that are from completely different strains. Normally we would need differential coverage across several samples to resolve this confidently, if at all possible.

Also at this level of similarity CheckM is not of much use as it measures the completeness and contamination based on the core-genome, hence you would never pick up potential errors from stitching together contigs from different strains as they share the core-genome (as it is highly likely to be co-assembled).

It is an open question how we best represent MAGs. I'm not sure if the best representation is the entire graph (figure 1A) or your walk through the graph. I think this should also be discussed in the paper.

Major comment 2: Binning

None of the bidders you tested were optimized for single-sample binning. Most would assume multiple coverage profiles, especially VAMP is highly dependent on high sample numbers for coverage profiles and assembly redundancy. This would be important to note in the paper.

With your relative simple binning implementation compared to other state-of-the-art software I would recommend most users to use your circle-rescue feature (if major comment 1 can be addressed), but afterwards apply a set of state-of-the-art tools and then dereplicate the recovered MAGs in the end - this has been standard in the field for some years now. Hence, what you should quantify is not how many bins each binning software adds, but how many unique ones they add. As you write, you have made a simple binning implementation, and if you searched more comprehensively in the literature I'm sure you would find something very close to your exact implementation.

Specific comments:

p1-57: "They could not reconstruct circular genomes automatically." This is just a fact of repeat content (internal or strains). Hence, in some cases it has been possible, although they are rare.. I think 1 additional line stating what the theoretical limitation is (repeats) would be warranted here for context.

p1-62: There are other completeness estimators than CheckM. Hence, you could instead write that CheckM is a commonly used tool to measure completeness.

p2-5: "near-complete" is not a term in the MiMAG standard. I'm not sure what you refer to here? The quality requirement is highly dependent on the question being asked, but in general Medium- or High-quality genomes are required.

p3-57: I would rarely recommend to rely on 1 sample coverage + 5-mer frequency for binning alone. Having more samples for differential coverage binning is almost always superior. Is that something that can be done in your framework?

p4-30: "extract contig bins in the embedded space using a small radius" -> the cluster identification is an important step, could you add a few more details in the main text to how this is done?

p4-45: "can recover a third to a fourth of HiFi MAGs in tested samples." What are you comparing to here?

p5-46: Please use the official MIMAG (not MISAG) quality definitions. I'm not sure where you found the ones you quoted, but they are not from the MIMAG paper you cited (see Tabel 1 in the paper you cite).

p5-58-61: "This heuristic" -> what does that refer to? In general the whole sentence is difficult to understand. What do you mean by "improved the representation of bacterial contents?".

p6-47: "by" should be "but".

p7-32: How did you reach this conclusion "SemiBin gives the most number of near-complete MAGs overall, mostly because it uses the same information as CheckM during binning." I would assume that is speculations unless you have tested it? SemiBin uses several different types of information that also would contribute, fx. taxonomic based constrains. However, I agree single-copy genes would likely have the largest impact on contamination.

Tabell: I do not understand what column 2 represent? Is that the "circle rescue"? But what is "circular complete contigs", "complete contigs" and "high-quality contigs"?

p9-7-11: The first sentences are very hard to understand. Please rewrite.

p9: Multiple references to figures are missing.

p11-61: How did you conclude that it is more expensive to generate MAGs with HiFi compared to short reads? That is not my experience, hence I would like to see the calculations to support this claim.

p12-31: "Major species" what is that?

p12-44-47: Using large sample catalogues and "hoping" for the species being present in decent abundance with low micro-diversity is often a very cost-effective strategy to recover genomes. It seems like you are questioning this strategy? Why is that? I think it is a completely fair way to do it.

p11-60 to p13-30: I do not understand this part of the discussion where you are comparing with the large short-read human gut datasets. First of all, it is a comprehensive analysis you have conducted and should be in the results section. Furthermore, I do not understand what you want to say with it. Ofcourse long-read PacBio sequencing is superior to short-read Illumina sequencing. Everyone knows that. I would skip this comparison altogether, to me it just blurs the aim of your paper and conclusions.

p13-39: What do you mean by "genome complete" and "representation complete" it is not phrases you have used before. Do you mean that all MAGs are complete and that they represent the entire sequenced metagenome? To me, there is a long way to go as we need to capture the strain-diversity. You have just captured, maybe, the most abundant strain in each MAG.

p13-44: What do you mean by the problem with reliance on CheckM?

Figure 5: Needs to be better annotated on the plot. Currently you need to read both the main text and the figure text to have a chance to understand it. Just add more text to the figure. Furthermore, while I intuitively think I understand the plot, I get unsure due to the "k-mer right-accumulated assembly multiplicity" mentioned on the y-axis. Would it be possible to simply visualize the fraction of read-kmers at a specific depth (x-axis) that are included in the assembly? That is what I intuitively thought the plots showed, but it seems not to be the case. Maybe a much more annotated supplementary figure is needed, unless there is something simple I'm overlooking.

Reviewer #1: I commend the authors on a much-improved version of their manuscript. I thoroughly enjoyed this reading. I also appreciate the detailed responses in the author's rebuttal, which I found convincing and well supported. I have only minor concerns with respect to the current version of the manuscript:

Minor issues

Q2.1.1. In P4, L27, I believe the authors meant to write that "512 is the total number of canonical 5-mers". Of course, 512 is also the number of non-canonical 5-mers, since 5 is an odd number and 5-mers cannot be identical to their reverse-complement, so the phrase is technically correct. However, I see no reason to confuse the reader with this.

A2.1.1 Thank you for pointing out the typo. We have corrected this in the revision.

Q2.1.2. In P9, L16 and throughout the manuscript the authors have left references to "Figure ??". Is there any information missing from the submission?

A2.1.2 There was a problem with our LaTeX source. When compiled with the journal's LaTeX system, it failed to show reference to supplementary figures. We are sorry for this error. We have edited the tex file to fix the issue.

Q2.1.3. In P9, L21, the authors write "within genome-wide read coverage range". It wasn't clear to me what this meant. Does this refer to the distribution of sequencing breadths of all genomes in the metagenome? Please clarify.

A2.1.3 By "within genome-wide read coverage range", we meant k-mers wouldn't be shown in Figure 5 if their counts are higher than the genome with the highest coverage. We have rewritten the paragraph that describes Figure 5 and rephrased the quoted sentence to: "*Given a metagenome sample composed of genomes with distinct sequences, a complete non-redundant assembly representing the entire sample will ideally have the 1× band spanning the entire plot up to the read depth of the most abundant genome.*"

Q2.1.4. The discussion of the manuscript retains a large portion of previously unreported results (P12, L36 to P13, L30). This is a matter of preference, but I believe it would be clearer for readers if this fragment of text was transferred to its own sub-section in Results, and only briefly discussed in the Discussion section.

A2.1.4 We moved the comparison of assemblies to catalogs into a new subsection in results and removed the reports about short read assembly. A similar question was also raised by the other reviewer.

Q2.1.5. In P13, L40ff, the authors write "..., bringing MAGs closer to their original goal which is to delegate their biosample with minimum bias". I disagree with this statement. The earliest attempts at genome-resolved metagenomics intended to retrieve genomes from some populations, but not to do community ecology from these genome catalogues. Indeed, even

today, the large majority of studies using MAGs address either aspects of population ecology or other species-specific aspects such as microdiversity, metabolic potential, distribution, etc. This could be an aim of some studies (and a very worthy aim!) but I don't think this was the "original goal".

A2.1.5 We have deleted this sentence and rewritten the entire paragraph.

Q2.1.6. In P14, L62 and elsewhere in the manuscript, the authors use the phrase "at least roughly 500Kb" or similar. Is there a reason why an exact number cannot be given instead? Is the algorithm adaptively changing that threshold? If so, how is this done? Or, if the authors mean that it's an implementation parameter that can be changed, with a default value of 500Kb, I would urge the authors to indicate this explicitly instead.

In P15, L6, the authors write "contains a non-significant amount of contigs". (1)What is that amount? Is it a specific fraction of the contigs in the DFS? Is it a set number of contigs? (2)How is this "significant amount" determined?

In P15, L10ff the authors explain when contigs are double-counted, but it wasn't clear to me what the "and so on so forth" is meant to indicate here. How is the pattern continued?

A2.1.6 We have rewritten the paragraph involving the three points to:

"We apply a depth-first search (DFS) to identify circular paths in the assembly graph. DFS discovers one circular path at a time. We keep a path if it contains ≥ 2 contigs and the path length is ≥ 500 kb. Here the path length is approximate as we ignore overlap lengths between contigs due to performance optimization. Furthermore, note that a contig may be used in many circular paths which may have similar sequences. To avoid visiting all these similar paths, at a fork in the graph we prioritize on the contig that has been used less often in previously found circular paths; we also reject a circular path if contigs in the path have been included for ≥ 100 times, in total, in previously found paths. With this threshold, we reduce paths sharing many contigs but there may still be redundant circular paths."

We hope this answers the reviewer's questions.

Q2.1.7 Additionally, I would recommend the authors revise the grammar of the manuscript thoroughly. I am convinced that this does not deter from the high scientific quality of the paper, but addressing some issues could facilitate the reading of the manuscript. During my own reading I collected a few suggestions, but this is not to be considered a comprehensive language revision:

1. P2, L28: "as whole" should be "as a whole"
2. P2, L33: "tempted" should be "tempting"
3. P2, L47: "abundance" should be "abundant"
4. P3, L48: "... (DFS) on the assembly..." should be "... (DFS) traversals on the assembly..."
5. P3, L52: "removes" should be "remove"

6. P6, L47: "by" should be "but"
7. P7, L14: "rest" should be "remaining"
8. P9, L12: "resulted" should be "resulting"
9. P15, L34: "such" should be "it"
10. P15, L54: "staring" should be "starting"

A2.1.7 We are grateful to the reviewer for identifying these errors. We have corrected them.

Reviewer #2: The manuscript "Evaluating and improving the representation of bacterial contents in long-read metagenome assemblies" by Feng and Li describes a new addition to hifiasm-meta that enable recovery of dominant genomes under strain-diversity. Furthermore, they describe a visualization using kmers to analyse what abundance-graded fraction of the metagenome that have assembled. I found the overall approaches highly interesting, both the genome binning in light of strain diversity and the abundance based visualization of recovery rate of MAGs. I previously reviewed the manuscript and it is now much easier to understand what the authors are trying to do.

Q2.2.1 Major comment 1: How to resolve strains I would like a more in depth introduction, results and discussion on the "rescue circle" part of the paper. To me, this is where most of the scientific contribution of this paper lies. As I understand, you go through the assembly graph and try to recover circular genomes. In the presence of strain diversity or extensive HGT (not that large a problem in long-read data) the assembly graph forks, as nicely showed in your figure 1A.

In the simplest case you have 2 strains and the forks then represents SNP's or entire genes that are different between the two strains. What is very unclear to me is how you take one or another path in the graph. It is stated that you do a "Depth-first search (DFS)" but you do not describe what is? Do you always take the path with the highest coverage through the graph? And how do you handle examples with strains in similar abundance, or when many strains exists? I'm a little concerned that the algorithm just put contigs together that are from completely different strains. Normally we would need differential coverage across several samples to resolve this confidently, if at all possible.

A.2.2.1 We have rewritten the paragraph that describes DFS to:

"We apply a depth-first search (DFS) to identify circular paths in the assembly graph. DFS discovers one circular path at a time. We keep a path if it contains ≥ 2 contigs and the path length is ≥ 500 kb. Here the path length is approximate as we ignore overlap lengths between contigs due to performance optimization. Furthermore, note that a contig may be used in many circular paths which may have similar sequences. To avoid visiting all these similar paths, at a fork in the graph we prioritize on the contig that has been used less often in previously found circular paths (ties are arbitrarily broken); we also reject a circular path if contigs in the path have been included for ≥ 100 times, in total, in previously found paths. With this threshold, we reduce paths sharing many contigs but there may still be redundant circular paths."

This describes how we choose contigs at a fork in the assembly graph. Note that at the circle finding stage, many contigs are composed of a few reads (the left figure in Figure 1A) and comparing coverages between such contigs is not reliable. Therefore we do not choose the contig with the highest coverage.

When there are multiple close strains tangled together, which strain is chosen is determined by the order of contig traversal. The result may be arbitrary. We note that “completely different strains” are likely to be separated already during contig assembly and won't be affected by circle finding. As a matter of fact, hifiasm-meta has higher power to separate strains than metaFlye, the only other published long-read metagenome assembler. Meanwhile, similar strains will be mixed by metaFlye as well, even more so with traditional binning algorithms. Our algorithm is not doing worse. In general, strain resolution is hard. Looking at assembly graphs such as the left figure in Fig 1A, we are not sure these tangled strains can be resolved with the current data.

Q2.2.2 Also at this level of similarity CheckM is not of much use as it measures the completeness and contamination based on the core-genome, hence you would never pick up potential errors from stitching together contigs from different strains as they share the core-genome (as it is highly likely to be co-assembled).

A2.2.2 We agree that checkM cannot evaluate contaminations across strains but CheckM is still useful to check incomplete MAGs and species-level contaminations.

Q2.2.3 It is an open question how we best represent MAGs. I'm not sure if the best representation is the entire graph (figure 1A) or your walk through the graph. I think this should also be discussed in the paper.

A2.2.3 Without circle findings, we would only get fragmented contigs, not MAGs from examples in Figure 1A. We now acknowledge this in Discussions: “*We showed that our algorithm led to more complete representation of bacterial populations in metagenome samples instead of fragmented contigs as shown in Figure 1A.*”

Major comment 2: Binning

Q2.2.4 None of the binners you tested were optimized for single-sample binning. Most would assume multiple coverage profiles, especially VAMP is highly dependent on high sample numbers for coverage profiles and assembly redundancy. This would be important to note in the paper.

A2.2.4 We thank the reviewer and now mention “*We also note that some binners, such as vamb, are optimized for jointly binning multiple samples and may underperform given a single sample.*”

Q2.2.5 With your relative simple binning implementation compared to other state-of-the-art software I would recommend most users to use your circle-rescue feature (if major comment 1 can be addressed), but afterwards apply a set of state-of-the-art tools and then dereplicate the

recovered MAGs in the end - this has been standard in the field for some years now. Hence, what you should quantify is not how many bins each binning software adds, but how many unique ones they add. As you write, you have made a simple binning implementation, and if you searched more comprehensively in the literature I'm sure you would find something very close to your exact implementation.

A2.2.5 We agree with the reviewer. We added a new column to Table 1. It shows the number of MAGs that can only be found by our hmBin algorithm but not by other binners in the table.

Specific comments:

Q2.2.6 p1-57: "They could not reconstruct circular genomes automatically." This is just a fact of repeat content (internal or strains). Hence, in some cases it has been possible, although they are rare.. I think 1 additional line stating what the theoretical limitation is (repeats) would be warranted here for context.

A2.2.6 We changed that sentence to "*They could not reconstruct circular genomes automatically due to repeat contents or strain similarity.*"

Q2.2.7 p1-62: There are other completeness estimators than CheckM. Hence, you could instead write that CheckM is a commonly used tool to measure completeness.

A2.2.7 We now say "*Completeness of a MAG is often measured with CheckM*", implying the possibility of other tools.

Q2.2.8 p2-5: "near-complete" is not a term in the MiMAG standard. I'm not sure what you refer to here? The quality requirement is highly dependent on the question being asked, but in general Medium- or High-quality genomes are required.

A2.2.8 The reviewer is correct that "near-complete" is not a classification used in MiMAG. However, we are unable to use MiMAG's exact definition anyway. In MiMAG, "high-quality draft" and lower quality brackets all specified that the MAG contains "multiple fragments" but we often see a MAG composed of one contig in our assembly. In addition, MiMAG's "finished" category is apparently defined for SAG, not for MAG. It requires Q50, which is hard to measure for individual MAGs without additional data. Also, as CheckM is reference-based and only gives approximate completeness, we cannot easily tell whether a MAG is really complete, either.

In the manuscript, we instead tried to follow conventions of SR metagenome assembly or catalog projects, where MiMAG principles are used, but "near-complete" is often the chosen phrase for MAGs that are highly complete ($\geq 90\%$), contain minimum contamination (usually $< 5\%$ or $< 10\%$), regardless of how many contigs does the MAG in question contain. We defined "near-complete" and other categories in our context, and we believe there is little ambiguity.

Q2.2.9 p3-57: I would rarely recommend to rely on 1 sample coverage + 5-mer frequency for

binning alone. Having more samples for differential coverage binning is almost always superior. Is that something that can be done in your framework?

A2.2.9 We agree with the reviewer. Unfortunately, most of the HiFi datasets were sequenced with PacBio Sequel I or II and were costly to produce. We do not have enough data to evaluate co-binning with multiple samples and therefore did not implement such algorithms.

Q2.2.10 p4-30: "extract contig bins in the embedded space using a small radius" -> the cluster identification is an important step, could you add a few more details in the main text to how this is done?

A2.2.10 We added the following description: "*We normalize each column in the matrix by Z-score, embed it to a 2-dimensional space with t-SNE and extract contig bins in the embedded space using a small radius: we seed from the longest non-circular contigs to the shortest. For each seed, we try to find a circle with diameter D on the plane such that it contains the seed and all other contigs that are $<D$ away from the seed ('neighbors'). If found, the seed and neighbors are put into a bin and marked as used; otherwise, we test whether the circle centered at the seed with diameter $1.6*D$ contains all 'neighbors', and create a bin on success. When both attempts fail, we record the seed as a single-contig bin if it is longer than 500Kb, otherwise do nothing.*"

Q2.2.11 p4-45: "can recover a third to a fourth of HiFi MAGs in tested samples." What are you comparing to here?

A2.2.11 We have changed the sentence to "*can recover a third to a fourth of near-complete MAGs found by hifiasm-meta in tested samples*".

Q2.2.12 p5-46: Please use the official MIMAG (not MISAG) quality definitions. I'm not sure where you found the ones you quoted, but they are not from the MIMAG paper you cited (see Tabel 1 in the paper you cite).

A2.2.12 We apologize for the typo and have corrected it. Please see **A2.2.8** about the terminology.

Q2.2.13 p5-58-61: "This heuristic" -> what does that refer to? In general the whole sentence is difficult to understand. What do you mean by "improved the representation of bacterial contents?"

A2.2.13 We clarified "this heuristic" to be "hmBin" which is defined in earlier texts. We removed the sentence of "improved the representation of bacterial contents" as it is redundant.

Q2.2.14 p6-47: "by" should be "but".

Q2.2.14 Thank you for pointing out this typo. We have made the change.

Q2.2.15 p7-32: How did you reach this conclusion "SemiBin gives the most number of near-complete MAGs overall, mostly because it uses the same information as CheckM during binning." I would assume that is speculation unless you have tested it? SemiBin uses several different types of information that also would contribute, fx. taxonomic based constrains. However, I agree single-copy genes would likely have the largest impact on contamination.

Q2.2.15 We have weakened the phrasing to: "*SemiBin gives the most number of near-complete MAGs overall, **potentially** because it uses the same information as CheckM during binning.*"

Q2.2.16 Tabel1: I do not understand what column 2 represents? Is that the "circle rescue"? But what is "circular complete contigs", "complete contigs" and "high-quality contigs"?

A2.2.16 In the revision, we put the thresholds of "near-complete" and "high-quality" MAGs in the table legend. These thresholds are also shown towards the end of Page 4. Take "74|78|17" of chicken-gut-1 for example. It means the assembly of this sample has 74 circular near-complete contigs, 78 linear or circular near-complete contigs, and 17 linear or circular high-quality contigs. Note that these three numbers evaluate the contig quality. The following columns evaluate the MAG quality.

Q2.2.17 p9-7-11: The first sentences are very hard to understand. Please rewrite.

A2.2.17 We have rephrased the sentence to "*Inspired by KAT and mercury, we use k-mers to evaluate the completeness and redundancy of a metagenome assembly.*"

Q2.2.18 p9: Multiple references to figures are missing.

A2.2.18 All references to supplementary figures were missing due to a problem with the LaTeX source. We apologize for the inconvenience and have fixed the issue in the new version. See also

Q2.2.19 p11-61: How did you conclude that it is more expensive to generate MAGs with HiFi compared to short reads? That is not my experience, hence I would like to see the calculations to support this claim.

p12-31: "Major species" what is that?

p12-44-47: Using large sample catalogs and "hoping" for the species being present in decent abundance with low micro-diversity is often a very cost-effective strategy to recover genomes. It seems like you are questioning this strategy? Why is that? I think it is a completely fair way to do it.

p11-60 to p13-30: I do not understand this part of the discussion where you are comparing with

the large short-read human gut datasets. First of all, it is a comprehensive analysis you have conducted and should be in the results section. Furthermore, I do not understand what you want to say with it. Of course long-read PacBio sequencing is superior to short-read Illumina sequencing. Everyone knows that. I would skip this comparison altogether, to me it just blurs the aim of your paper and conclusions.

A2.2.19 We agree with the reviewer. We have removed one paragraph from p11-60 to p13-8 in the previous version of the manuscript. Questions related to p11-61, p12-31 and p12-44-47 are not applicable now. We moved the paragraph at p13-9-30 to the last subsection in Results. This subsection shows that the assembly of a small number of HiFi datasets can find MAGs not present in metagenome catalogs made from many short-read samples. Quantifying how much HiFi assembly contributes is still useful information to general readers.

Q2.2.20 p13-39: What do you mean by "genome complete" and "representation complete" it is not phrases you have used before. Do you mean that all MAGs are complete and that they represent the entire sequenced metagenome? To me, there is a long way to go as we need to capture the strain-diversity. You have just captured, maybe, the most abundant strain in each MAG.

p13-44: What do you mean by the problem with reliance on CheckM?

A2.2.20 By "representation complete", we meant we could assemble species with enough read coverage. We don't expect to assemble species at coverage of a few folds. At p13-44, we meant CheckM is reference-based and may underestimate the completeness not well represented in its reference dataset. We have rewritten the paragraph to address these concerns:

"We also described two reference-free approaches, k-mer spectrum and species-level OTUs based on full length 16S rRNAs, to evaluate how well prevalent species in a metagenome sample are represented by a metagenome assembly. Unlike CheckM, our methods do not depend on known genomes or single-copy core genes and thus are not biased towards existing reference genomes. Applying the methods to real data, we showed that de novo HiFi assembly plus binning can sometimes assemble the great majority of prevalent species into near-complete MAGs with many of them not seen in the existing metagenome catalogs produced from short reads."

Q2.2.21 Figure 5: Needs to be better annotated on the plot. Currently you need to read both the main text and the figure text to have a chance to understand it. Just add more text to the figure. Furthermore, while I intuitively think I understand the plot, I get unsure due to the "k-mer right-accumulated assembly multiplicity" mentioned on the y-axis. Would it be possible to simply visualize the fraction of read-kmers at a specific depth (x-axis) that are included in the assembly? That is what I intuitively thought the plots showed, but it seems not to be the case. Maybe a much more annotated supplementary figure is needed, unless there is something simple I'm overlooking.

A2.2.21 The Figure 5 legend gives the exact definition of the Y axis. To make this more explicit, we mentioned that $N_x^{(c)}$ is “right-accumulated” in the revision. We also pointed out that “*The height of the blue area ... is the fraction of read k-mers occurring $\geq x$ times in reads but absent from the assembly.*” This is similar to the reviewer’s interpretation except that it is “right-accumulated”. We use right-accumulation because the plot with marginal count is noisy and hard to read. We also rewrote the paragraph in the main text that describes Figure 5 and hope it is easier to understand now.

Third round of review

Reviewer 2

I have reviewed this paper several times, it has improved much and it's now easier to understand what the authors are trying to do.

As pointed out by me and the other reviewers several times, although the text is readable, it could benefit from a professional rewrite. The authors have nicely corrected the text-errors highlighted by the reviewers, but it seems like a larger effort to rewrite the text for clarity has not been conducted. The highlighted text errors were just a few examples.

P4-line48: You can not say you follow MiMAG standards and then simply define other categories (near-complete) and redefine the criteria for HQ. Also, your definition of near-complete is basically the same as the MIMAG HQ definition. It is really confusing. Please read the MIMAG paper and either use their standards (you can have a MIMAG HQ genome in 1 contig if it does not satisfy the finished criteria) or define your own. In the table text you sometimes still refer to "HQ contigs", not "HQ MAGs".

I'm still confused about how the "circle rescue" works. From the answer and manuscript, it seems that a random path in the graph is taken - as long as the algorithm can make something "circular". However, this will just create chimeric assemblies and is not useful. I would much rather have a bin in 100 pieces than a chimeric assembly in 1 piece. It seems like the authors are very focused on having "complete" genomes by measuring the completeness of core genes - and not if the assembly actually represents a true biological entity. I initially understood that the "Depth First Search" meant that the algorithm would transverse the graph taking a high-coverage path. In my mind that makes sense as in many cases it is low-abundant strain diversity that breaks the assembly. However, it seems to be different from how the authors describe the DFS in the manuscript and review response. If it just is a random walk through the graph I can not recommend other people to use it, it will simply create a flood of chimeric assemblies that will pollute the databases. The simplest example is graph 3 in Figure 1 - why did your algorithm take that specific path? It seems like multiple others could be just as likely if not other information is used?

My main worry is this; With highly fragmented bins it is easy to have lower confidence in the results. I'm very worried that "on paper" nicely looking single-contig and 100% complete assemblies - but chimeric! - will be impossible to remove from the databases later and blur numerous downstream analyses in the future.

Reviewer #2: I have reviewed this paper several times, it has improved much and it's now easier to understand what the authors are trying to do.

Q3.2.1 As pointed out by me and the other reviewers several times, although the text is readable, it could benefit from a professional rewrite. The authors have nicely corrected the text-errors highlighted by the reviewers, but it seems like a larger effort to rewrite the text for clarity has not been conducted. The highlighted text errors were just a few examples.

A3.2.1 We have rewritten several paragraphs in Results and in Methods. We hope this improves the quality of this manuscript.

Q3.2.2 P4-line48: You can not say you follow MiMAG standards and then simply define other categories (near-complete) and redefine the criteria for HQ. Also, your definition of near-complete is basically the same as the MIMAG HQ definition. It is really confusing. Please read the MIMAG paper and either use their standards (you can have a MIMAG HQ genome in 1 contig if it does not satisfy the finished criteria) or define your own. In the table text you sometimes still refer to "HQ contigs", not "HQ MAGs".

A3.2.2 In the last response A2.2.8 we have clarified that the original MiMAG standard cannot be applied to long-read MAGs literally. To avoid confusion, we do not mention MiMAG in the revised manuscript anymore. When describing our criteria, we only say "Following previous work [20–22]" which does not include the MiMAG paper.

Q3.2.3 I'm still confused about how the "circle rescue" works. From the answer and manuscript, it seems that a random path in the graph is taken - as long as the algorithm can make something "circular". However, this will just create chimeric assemblies and is not useful. I would much rather have a bin in 100 pieces than a chimeric assembly in 1 piece. It seems like the authors are very focused on having "complete" genomes by measuring the completeness of core genes - and not if the assembly actually represents a true biological entity. I initially understood that the "Depth First Search" meant that the algorithm would transverse the graph taking a high-coverage path. In my mind that makes sense as in many cases it is low-abundant strain diversity that breaks the assembly. However, it seems to be different from how the authors describe the DFS in the manuscript and review response. If it just is a random walk through the graph I can not recommend other people to use it, it will simply create a flood of chimeric assemblies that will pollute the databases. The simplest example is graph 3 in Figure 1 - why did your algorithm take that specific path? It seems like multiple others could be just as likely if not other information is used?

My main worry is this; With highly fragmented bins it is easy to have lower confidence in the results. I'm very worried that "on paper" nicely looking single-contig and 100% complete assemblies - but chimeric! - will be impossible to remove from the databases later and blur numerous downstream analyses in the future.

A3.2.3 In the revision, we have clarified that hifiasm-meta kicks off DFS from the longest unvisited contig. Note that always choosing the contig of the highest coverage would be sensitive to HGT and transposons. It would not work well in practice.

As we explained in A2.2.1 in the last round of the review, no existing metagenome assemblers can distinguish closely related strains. Metagenome bidders are worse because the tetra-mer contents between two similar strains are indistinguishable and the read coverage often varies greatly depending

on the divergence and the strain diversity in a region. Many MAGs, in particular short-read MAGs, represent multiple strains, not single genomes. To this end, circular contigs found DFS are not worse than MAGs. Our evaluation suggested so, too. Note that circular contigs found by DFS are clearly labeled in the hifiasm-meta output. They are not treated the same as circular contigs.